# Functionally coupled ion channels begin co-assembling at the start of their synthesis

**Roya Pournejati[1,2], Jessica M Huang[1,2], Michael Ma[1,2], Claudia M Moreno[2,3], Oscar Vivas[1,2]***

[1]Department of Pharmacology, University of Washington, Seattle, United States; [2]Department of Neurobiology and Biophysics, University of Washington, Seattle, United States; [3]Howard Hughes Medical Institute, Chevy Chase, United States

## eLife Assessment

This **fundamental** manuscript provides **compelling** evidence that BK and CaV1.3 channels can co-localize as ensembles early in the biosynthetic pathway, including within the ER and Golgi. The findings, supported by a range of imaging and proximity assays, offer insights into channel organization in both heterologous and endogenous systems. The data substantiate the central claims, while highlighting intriguing mechanistic questions for future studies: the determinants of mRNA co-localization, the temporal dynamics of ensemble trafficking, and the physiological implications of pre-assembly for channel function at the plasma membrane.

**\*For correspondence:**
vivas@uw.edu

**Competing interest:** The authors declare that no competing interests exist.

## Abstract

Calcium binding to BK channels lowers BK activation threshold, substantiating functional coupling with calcium-permeable channels. This coupling requires close proximity between different channel types, and the formation of BK-Ca$_V$1.3 hetero-clusters at nanometer distances exemplifies this unique organization. To investigate the structural basis of this interaction, we tested the hypothesis that BK and Ca$_V$1.3 channels assemble before their insertion into the plasma membrane. Our approach incorporated four strategies: (1) detecting interactions between BK and Ca$_V$1.3 proteins inside the cell, (2) identifying membrane compartments where intracellular hetero-clusters reside, (3) measuring the proximity of their mRNAs, and (4) assessing protein interactions at the plasma membrane during early translation. These analyses revealed that a subset of BK and Ca$_V$1.3 transcripts are spatially close in micro-translational complexes, and their newly synthesized proteins associate within the endoplasmic reticulum (ER) and Golgi. Comparisons with other proteins, transcripts, and randomized localization models support the conclusion that BK and Ca$_V$1.3 hetero-clusters form before their insertion at the plasma membrane.

## Introduction

This work concerns the organization of functionally coupled voltage- and calcium-dependent potassium (BK) channels and voltage-gated calcium (Ca$_V$) channels.

BK channels (a.k.a. K$_{Ca}$1.1, Maxi-K, Slo1, KCNMA1) are named for their 'big potassium (K$^+$)' conductance, since channel opening leads to unusually large, outward potassium currents and membrane repolarization. Its amino acid sequence is conserved throughout the animal kingdom from worms to mammals. BK channels are expressed in a wide variety of cell types, predominantly excitable cells but also non-excitable salivary, bone, and kidney cells, making this channel responsible for a large range of physiological processes (*Contreras et al., 2013*; *Kaczorowski et al., 1996*; *Contet et al., 2016*;

*Ancatén-González et al., 2023*; *Dopico et al., 2018*; *Hu et al., 1991*; *Semenov et al., 2006*; *Dupont et al., 2024*; *Pallotta et al., 1981*). Phenotypes of pathological BK mutations in human patients are most prominent in the brain and muscle and often manifest as seizures, movement disorders, developmental delay, and intellectual disability (*Meredith, 2024*; *Bailey et al., 2019*). BK is also implicated in other organ system functions, including but not limited to cardiac pacemaking, reproduction, and pancreatic glucose homeostasis (*Bailey et al., 2019*; *Houamed et al., 2010*).

Since BK channels affect function in numerous physiological systems, channel opening is meticulously regulated at the cellular level. BK is a voltage-gated channel activated by membrane depolarization. Interestingly, BK has an additional gating mechanism that differentiates it from typical voltage-gated potassium channels; BK opening is gated by both voltage and intracellular calcium binding to a cytosolic regulatory domain. At the cytoplasmic resting free calcium concentration (~0.1 µM), BK channels remain closed (*Shah et al., 2021*). However, the probability of channel opening increases when both the membrane potential depolarizes and when the local free calcium rises. Yet, increases in calcium are tightly limited by endogenous proteins that bind calcium with high affinity, as well as by extrusion mechanisms that take calcium inside organelles or outside the cell. Hence, the activation of BK channels relies on strategies to overcome these regulatory barriers. One such strategy is to localize near sources of calcium.

In excitable cells, BK forms nanodomains with calcium channels that provide exclusive, localized calcium sources. Several subtypes of calcium channels form functional signaling complexes with BK, shifting BK activation voltages to more negative potentials (*Berkefeld et al., 2006*; *Vivas et al., 2017*; *Prakriya and Lingle, 2000*). One of these channels, $Ca_V1.3$ (a.k.a. *CACNA1D*), is unique in its electrophysiological profile as an L-type calcium channel, activating with fast kinetics at voltages as negative as –55 mV (*Lipscombe et al., 2004*). Additionally, BK channels are modulated by auxiliary subunits, which fine-tune BK channel gating properties to adapt to different physiological conditions. The β, γ, and LINGO1 subunits each contribute distinct structural and regulatory features: β-subunits modulate $Ca^{2+}$ sensitivity and can induce inactivation; γ-subunits shift voltage-dependent activation to more negative potentials; and LINGO1 reduces surface expression and promotes rapid inactivation. These interactions ensure precise control over channel activity, allowing BK channels to integrate voltage and calcium signals dynamically in various cell types (*Chen et al., 2023a*; *Brenner et al., 2000*; *Dudem et al., 2023*; *Dudem et al., 2020*; *Knaus et al., 1994*; *Yan and Aldrich, 2010*; *Yan and Aldrich, 2012*).

Here, we focus on the selective assembly of BK channels with $Ca_V1.3$ and do not evaluate the contributions of auxiliary subunits to BK channel organization. $Ca_V1.3$ is expressed in many of the same cell types as BK and is often functionally coupled with BK channels, enabling BK channels to activate at more negative voltages. Notably, super-resolution microscopy shows that $Ca_V1.3$ organizes spatially into nanodomains with BK in the plasma membrane (*Vivas et al., 2017*). However, the mechanisms behind the assembly of BK and $Ca_V1.3$ hetero-clusters remain unknown.

**Table 1.** List of mechanisms described for the interaction between channel subunits, channels of the same type (homo-clusters), or hetero-clusters of channel families permeating different ions.

| Mechanism | Description | Source |
|---|---|---|
| Co-translation of heteromeric channel subunits | mRNA transcripts and nascent proteins of hERG heteromeric subunits form molecular complexes during protein translation | *Liu et al., 2016* |
| Co-translation of channels permeating different ions | Potassium channel hERG and sodium channel SCN5A form complexes of mRNA transcripts and nascent proteins during protein translation | *Eichel et al., 2019* |
| Membrane curvature sensing | Clusters of Piezo1 channels enriched in membrane invaginations | *Yang et al., 2022* |
| ER membrane protein complex | ER membrane complex acts as a chaperone for heteromeric channel assembly | *Chen et al., 2023b* |
| Scaffolding proteins | Scaffolding protein AKAP150 is required for abnormal gating of $Ca_V1.2$-LQT8 channels | *Cheng et al., 2011* |
| Random insertion | Clusters of $Ca_V1.2$, $Ca_V1.3$, BK, and TRPV4 are proposed to be randomly formed into the plasma membranes of smooth muscle, cardiac muscle, hippocampal neurons, and tsA-201 cells | *Sato et al., 2019* |

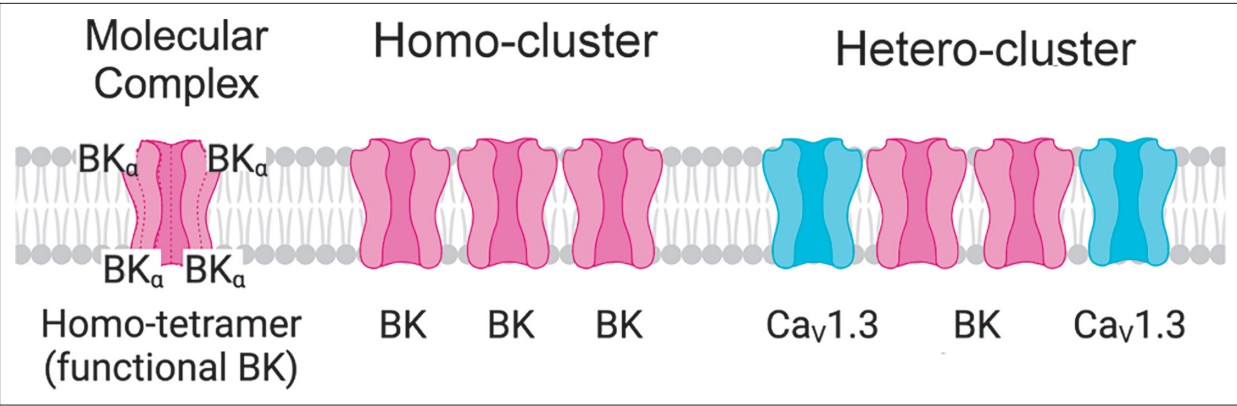

**Figure 1.** Representation of molecular complex, homo-cluster, and hetero-cluster.

Several mechanisms for bringing ion channels together have been suggested (*Table 1*). One proposes that protein assembly precedes protein insertion into the plasma membrane (*Liu et al., 2016*; *Eichel et al., 2019*; *Lu et al., 2001*). This mechanism has been observed in studies of hetero-multimers and even channels permeating different ions. The formation of these groups of proteins would start from their synthesis. There is even the possibility of mRNA transcripts colocalizing prior to translation. We explored this mechanism in relation to BK and Ca$_V$1.3 functional coupling. Here, we started by investigating when and where BK and Ca$_V$1.3 channels cluster in the cell. We looked for BK and Ca$_V$1.3 hetero-clusters at intracellular membranes of the ER, at ER exit sites, and at the Golgi. We also investigated the proximity between mRNA transcripts of BK and Ca$_V$1.3.

### Definitions used in this manuscript

To guide the reader and prevent confusion on the terms used, we introduce the following nomenclature, which is also illustrated in *Figure 1*. Molecular complex: an array of several polypeptides with a defined function. In our case, a BK channel is a molecular complex of four α-subunits whose function is to permeate ions. Homo-cluster: we define a homo-cluster as the accumulation of proteins. In our work, a homo-cluster refers to the accumulation of BK channels or the accumulation of calcium channels.

Hetero-cluster: we define a hetero-cluster as the collection of different types of proteins that facilitate functional coupling and compartmentalization of a signaling complex. In the present work, hetero-cluster refers to a coordinated collection of BK and Ca$_V$ channels. It is also used to refer to the assembly of homo-clusters of BK with homo-clusters of Ca$_V$ channels.

## Results

### BK and Ca$_V$1.3 hetero-clusters are found inside the cell

BK and Ca$_V$ hetero-clusters have been observed at the plasma membrane (*Berkefeld et al., 2006*; *Vivas et al., 2017*; *Prakriya and Lingle, 2000*). However, a clear understanding of when and how BK and Ca$_V$ hetero-clusters assemble is lacking. A simple mechanism proposes that these hetero-clusters organize only at the plasma membrane. Here, we tested the alternative hypothesis of BK and Ca$_V$ assembling inside the cell (*Figure 2A*). To test this idea, we used proximity ligation assay (PLA) and antibodies against BK and Ca$_V$1.3 channels. When these antibodies are within 40 nm of each other, PLA ligation and amplification can occur, resulting in the formation of fluorescent puncta (*Figure 2B*), here referred to as PLA puncta. Hence, the PLA puncta in *Figure 2C* represent BK and Ca$_V$1.3 hetero-clusters. To confirm specificity, a negative control was performed by probing only for BK using the primary antibody, ensuring that detected signals were not due to nonspecific binding or background fluorescence. We first analyzed Z-projections, in which all the puncta in the cell volume are added, and found a density of 5.8±1.0/10 µm². In a different experiment, we analyzed the puncta density for each focal plane of the cell (step size of 300 nm) and compared the puncta at the plasma membrane to the rest of the cell. We visualized the plasma membrane with a biological sensor tagged with GFP

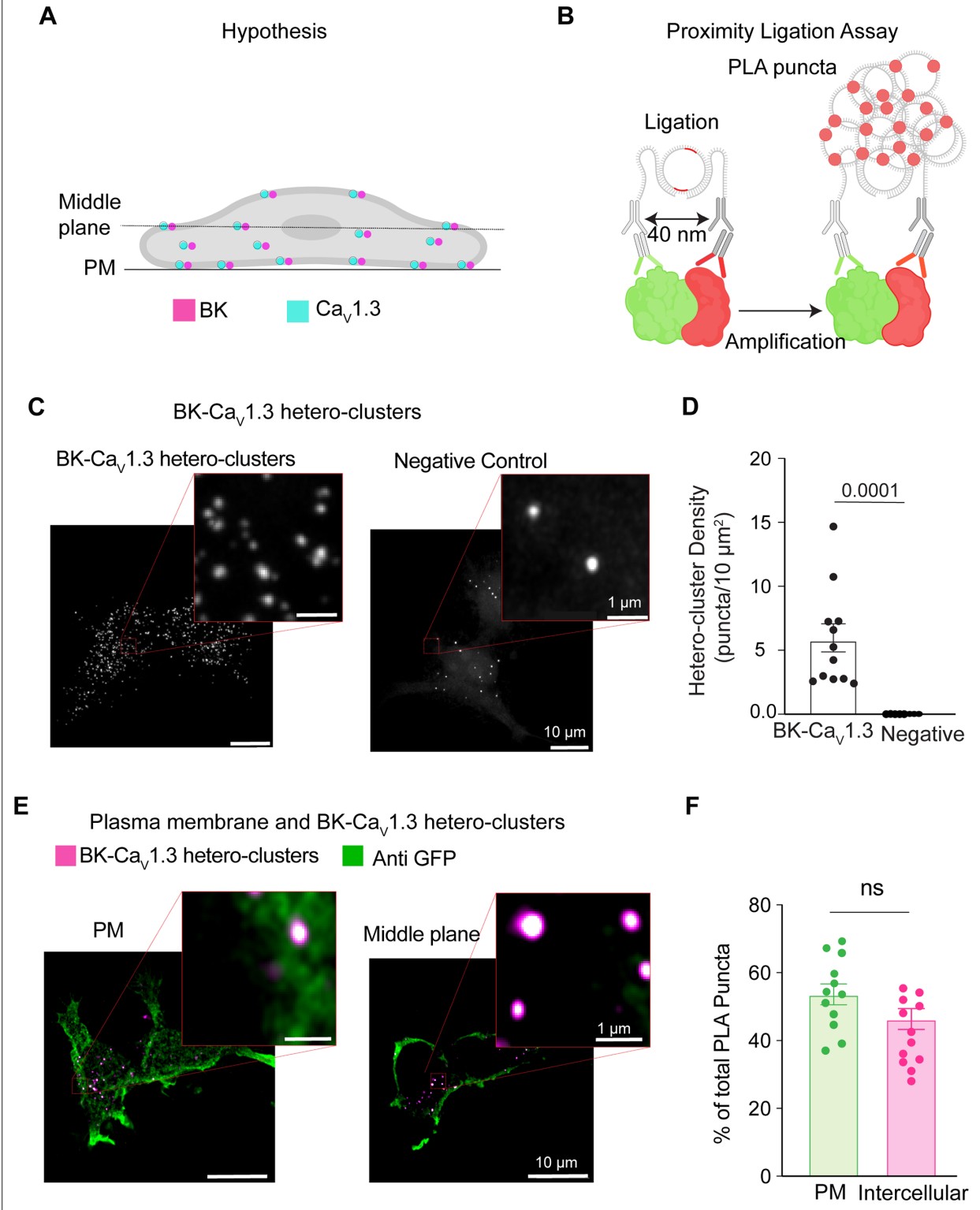

**Figure 2.** BK and Ca$_V$1.3 hetero-clusters are found inside the cell. (**A**) Diagram of the hypothesis: hetero-clusters of BK (magenta) and Ca$_V$1.3 (cyan) are on intracellular membranes and on the plasma membrane. (**B**) Illustration of the technique to detect BK and Ca$_V$1.3 hetero-clusters. Proximity ligation assay is used to detect the hetero-clusters. (**C**) Confocal images of fluorescent puncta from proximity ligation assay (PLA) experiments in tsA-201 cells. Left: Cells were transfected and probed for BK and Ca$_V$1.3 channels. Right: negative control. Cells were transfected and probed only for BK channels. Enlargement of a selected region is shown in the inset. (**D**) Scatter dot plot comparing puncta density of BK and Ca$_V$1.3 hetero-clusters to the negative control. Data points are from n=12 cells for BK and Ca$_V$1.3 hetero-clusters and from n=14 cells for negative control. p-Values are shown at the top of the

*Figure 2 continued on next page*

*Figure 2 continued*

graphs. (**E**) Confocal images of fluorescent PLA puncta at different focal planes co-labeled against GFP at the plasma membrane. Cells were transfected with BK, Ca$_V$1.3, and PH-PLCδ-GFP and probed for BK channels, Ca$_V$1.3 channels, and GFP. PLA puncta are shown in magenta, and the plasma membrane is shown in green. Enlargements of the representative regions of PM and intercellular hetero-clusters are shown in the insets. (**F**) Scatter dot plot comparing BK and Ca$_V$1.3 hetero-cluster abundance at PM and inside the cell. Data points are from n=12 cells. Scale bars are 10 μm and 1 μm in the insets. For statistical analysis, unpaired t-tests were applied in panel D and paired t-tests in panel F to evaluate significance.

The online version of this article includes the following figure supplement(s) for figure 2:

**Figure supplement 1.** Validation of antibodies against BK, Ca$_V$1.3, and GFP.

(PH-PLCδ-GFP) and then probed it with an antibody against GFP (*Figure 2E*). By analyzing the GFP signal, we created a mask that represented the plasma membrane. The mask served to distinguish between the PLA puncta located inside the cell and those at the plasma membrane, allowing us to calculate the number of PLA puncta at the plasma membrane. To our surprise, we found a significant number of puncta localized inside the cell. 46 ± 3% of the puncta were localized intracellularly, whereas 54 ± 3% were at the plasma membrane. This finding is consistent with our supposition that BK and Ca$_V$1.3 channels colocalize in the cell.

## BK and Ca$_V$1.3 hetero-clusters localize at ER and ER exit sites

We next investigated the identity of the intracellular membranes where these PLA puncta were found. A large component of intracellular membranes in the cell is the endoplasmic reticulum (ER), where channels are inserted after translation. To determine whether BK and Ca$_V$1.3 associate in the ER (*Figure 3A*), we combined PLA with immunodetection. As before, PLA probed for BK and Ca$_V$1.3 hetero-clusters, while a (*Contreras et al., 2013*) KDEL-moxGFP label identified the ER (*Figure 3B*). To avoid disruption of organelle architecture, we used the monomeric mox version of KDEL-GFP, which is optimized to reduce oligomerization in the ER environment (*Costantini et al., 2015*). Neither overexpression of KDEL-moxGFP nor fixation altered the ER structure as the tubule width remained around 150 nm for either condition (*Figure 3C*), a value in agreement with the literature (*Georgiades et al., 2017*; *Nixon-Abell et al., 2016*). To assess the percentage of PLA puncta colocalizing with the ER, we employed two different cell lines: our overexpression system (tsA-201 cells) and a rat insulinoma cell line (INS-1) that expresses BK and Ca$_V$1.3 channels endogenously. In this and following experiments, we analyze one focal plane, in the middle of the cell, to quantify PLA puncta colocalization with ER membrane. *Figure 3D* shows PLA puncta in the same space as the ER. Comparing the overexpression and endogenous systems, 63 ± 3% vs 50 ± 6% of total PLA puncta were localized at the ER (*Figure 3E*). To determine whether the observed colocalization between BK-Ca$_V$1.3 hetero-clusters and the ER was not simply due to the extensive spatial coverage of ER labeling, we labeled ER exit sites using Sec16-GFP and probed for hetero-clusters with PLA. This approach enabled us to test whether the hetero-clusters were preferentially localized to ER exit sites, which are specialized trafficking hubs that mediate cargo selection and direct proteins from the ER into the secretory pathway. In contrast to the more expansive ER network, which supports protein synthesis and folding, ER exit sites ensure efficient and selective export of proteins to their target destinations. By quantifying the proportion of BK and Ca$_V$1.3 hetero-clusters relative to total channel expression at ER exit sites, we found 28 ± 3% colocalization in tsA-201 cells and 11 ± 2% in INS-1 cells (*Figure 3F*). While the percentage of colocalization between hetero-clusters and the ER or ER exit sites alone cannot be directly compared to infer trafficking dynamics, these findings reinforce the conclusion that hetero-clusters reside within the ER and suggest that BK and Ca$_V$1.3 channels traffic together through the ER and exit in coordination.

## BK and Ca$_V$1.3 hetero-clusters go through the Golgi

Channels are modified in the Golgi after synthesis (*Figure 4A*), so we asked whether PLA puncta formed by BK and Ca$_V$1.3 hetero-clusters can be found in the Golgi. Using the same strategy, we labeled Golgi with Gal-T-mEGFP (*Figure 4B*) and detected BK-Ca$_V$1.3 hetero-clusters using PLA. To confirm that the overexpressed Gal-T-mEGFP labels the Golgi specifically without altering its structure, we compared the region detected by the antibody against GFP to the region detected by a primary antibody against the Golgi protein 58K (*Figure 4C*). 58K is a specific peripheral protein that localizes to the cytosolic face of Golgi (*Bashour and Bloom, 1998*). The overlay shows that the GFP

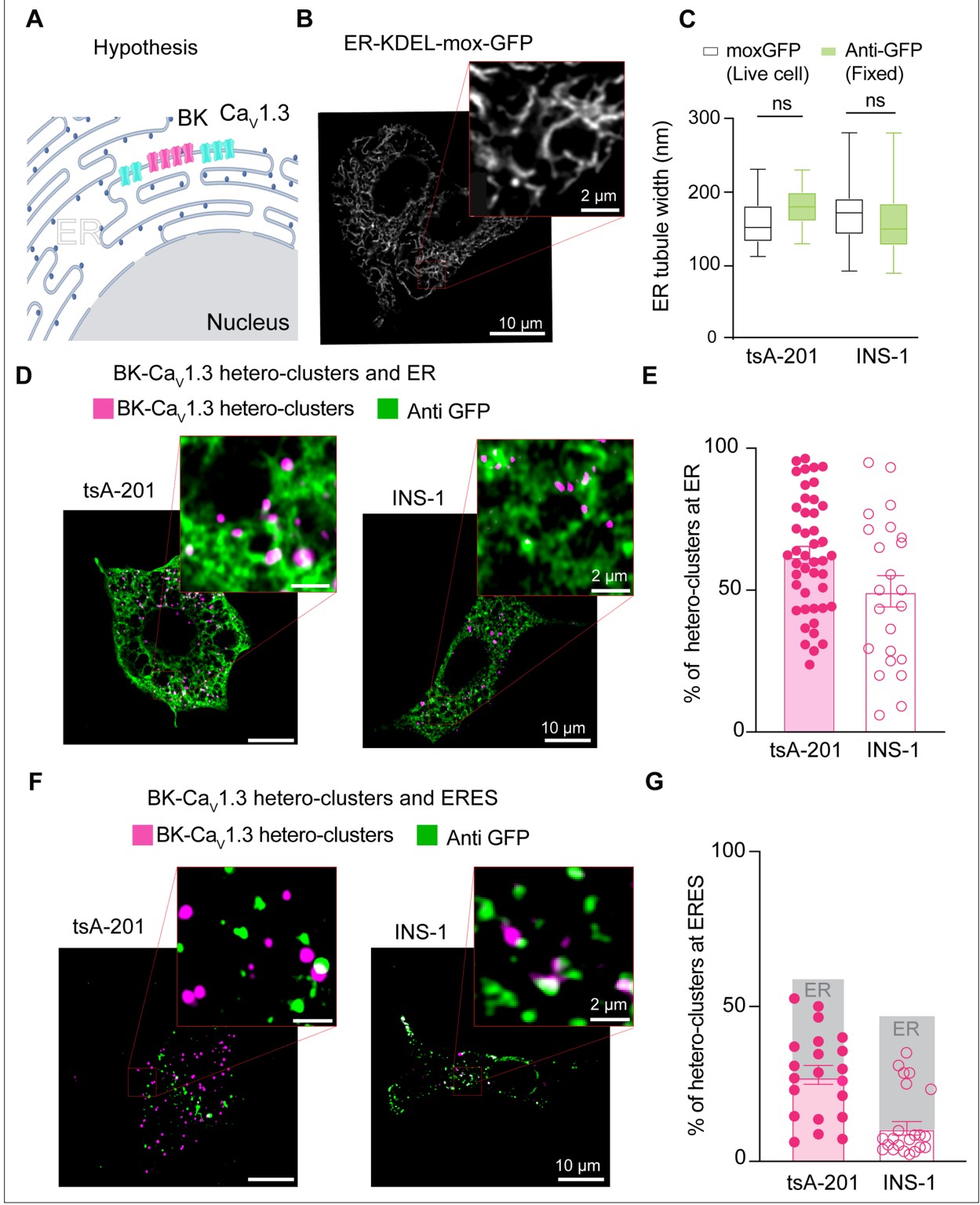

**Figure 3.** BK and Ca$_V$1.3 hetero-clusters localize at endoplasmic reticulum (ER) and ER exit sites (ERES). (**A**) Diagram of the hypothesis: hetero-clusters of BK (magenta) and Ca$_V$1.3 (cyan) can be found at the ER membrane. (**B**) Representative image of the ER labeled with exogenous GFP in INS-1 cells. Cells were transfected with KDEL-moxGFP. Magnification is shown in the inset. (**C**) Comparison of the ER tubule distance in live and fixed tsA-201 and INS-1 cells. Data points are from n=23 tsA-201 cells, n=27 INS-1 cells. (**D**) Representative images of proximity ligation assay (PLA) puncta and ER. Left: tsA-201 cells were transfected with BK, Ca$_V$1.3, and KDEL-moxGFP. Right: INS-1 cells were transfected only with KDEL-moxGFP. Fixed cells were probed for BK-Ca$_V$1.3 hetero-clusters (PLA puncta) and GFP. PLA puncta are shown in magenta. ER is shown in green. (**E**) Comparison of BK-Ca$_V$1.3 hetero-clusters

*Figure 3 continued on next page*

*Figure 3 continued*

found at the ER and relative to all PLA puncta in the cell. Values are given in percentages. (**F**) Representative images of PLA puncta and ERES. Left and right are the same as in D, but cells were transfected with Sec16-GFP instead of KDEL. (**G**) Comparison of BK-Ca$_V$1.3 hetero-clusters found at ERES relative to all PLA puncta in the cell. Values are given in percentages. Data points are from n=45 tsA-201 cells for ER, n=21 tsA-201 cells for ERES, n=23 INS-1 cells for ER, and n=23 INS-1 cells for ERES. Scale bars are 10 μm and 2 μm in the magnifications.

signal labels the same region as the antibody against 58K. When assessing the presence of BK-Ca$_V$1.3 hetero-clusters in the Golgi, we found that 31 ± 5% of PLA puncta were localized at the Golgi in the overexpression system and 25 ± 4% in the endogenous cell model (*Figure 4E*).

We performed controls to confirm that the formation of hetero-clusters between the two channels was not coincidental but rather the result of structural coupling. We tested the formation of PLA puncta between BK channels and the Golgi protein 58K. We also tested for Ca$_V$1.3 and 58K. We did not find PLA puncta when the proximity between the channels and 58K was probed (*Figure 4F*), supporting the idea that PLA puncta between BK and Ca$_V$1.3 channels found at the Golgi represent specific coupling. We selected the Golgi as a control because it represents the final stage of protein trafficking, ensuring that hetero-cluster interactions observed at this point reflect specificity maintained throughout earlier trafficking steps, including within the ER. As an additional control, we tested the formation of PLA puncta between Ca$_V$1.3 channels and RyR$_2$, a protein localized in the ER. The number of PLA puncta between Ca$_V$1.3 and RyR$_2$ was significantly lower than the number observed between Ca$_V$1.3 and BK channels (*Figure 4—figure supplement 1*), further supporting the specificity of BK-Ca$_V$1.3 interactions.

It is important to clarify that the percentages provided in this work cannot be added up to understand the distribution of hetero-clusters along the biosynthetic pathway of the channels. The percentage in each membrane compartment was compared to the total percentage of hetero-clusters observed in each cell for that particular experiment. Therefore, there is expected overlap between our measurements due to (1) optical resolution and (2) the effect of not comparing all the organelles in the same cell. Considering these limitations, we interpreted our results as follows: about one half of hetero-clusters are found inside the cell. The other half corresponds to hetero-clusters at the plasma membrane. From the half inside the cell, roughly one third of hetero-clusters is found in the Golgi, and another third is found in ER exit sites (*Figure 4G*). The remaining hetero-clusters are found in other regions of the ER and in membranes that we did not probe, such as vesicles. Finally, a key limitation of this approach is that we cannot quantify the proportion of total BK or Ca$_V$1.3 channels engaged in hetero-clusters within each compartment. The PLA method provides proximity-based detection, which reflects relative localization rather than absolute channel abundance within individual organelles.

## BK mRNA and Ca$_V$1.3 mRNA colocalize

How is it that BK and Ca$_V$1.3 proteins come in close proximity in membranes of the ER, ER exit sites, and the Golgi? To explore the origins of the initial association, we hypothesized that the two proteins are translated near each other, which could be detected as the colocalization of their mRNAs (*Figure 5A and B*). The experiment was designed to detect single mRNA molecules from INS-1 cells in culture. We performed multiplex in situ hybridization experiments using an RNAscope fluorescence detection kit to be able to image three mRNAs simultaneously in the same cell and acquired the images in a confocal microscope with high resolution. To rigorously assess the specificity of this potential mRNA-level organization, we used multiple internal controls. *GAPDH* mRNA, a highly expressed housekeeping gene with no known spatial coordination with channel mRNAs, served as a baseline control for nonspecific colocalization due to transcript abundance. To evaluate whether the spatial proximity between BK mRNA (*KCNMA1*) and Ca$_V$1.3 mRNA (*CACNA1D*) was unique to functionally coupled channels, we also tested for Na$_V$1.7 mRNA (*SCN9A*), a transmembrane sodium channel expressed in INS-1 cells but not functionally associated with BK. This allowed us to determine whether the observed colocalization reflected a specific biological relationship rather than shared expression context. Finally, to test whether this proximity might extend to other calcium sources relevant to BK activation, we probed the mRNA of ryanodine receptor 2 (*RyR2*), another Ca$^{2+}$ channel known to interact structurally with BK channels (*Lifshitz et al., 2011*). Together, these controls were chosen to distinguish specific mRNA colocalization patterns from random spatial proximity, shared subcellular distribution, or gene expression level artifacts. *Figure 5—figure supplement 2* shows images

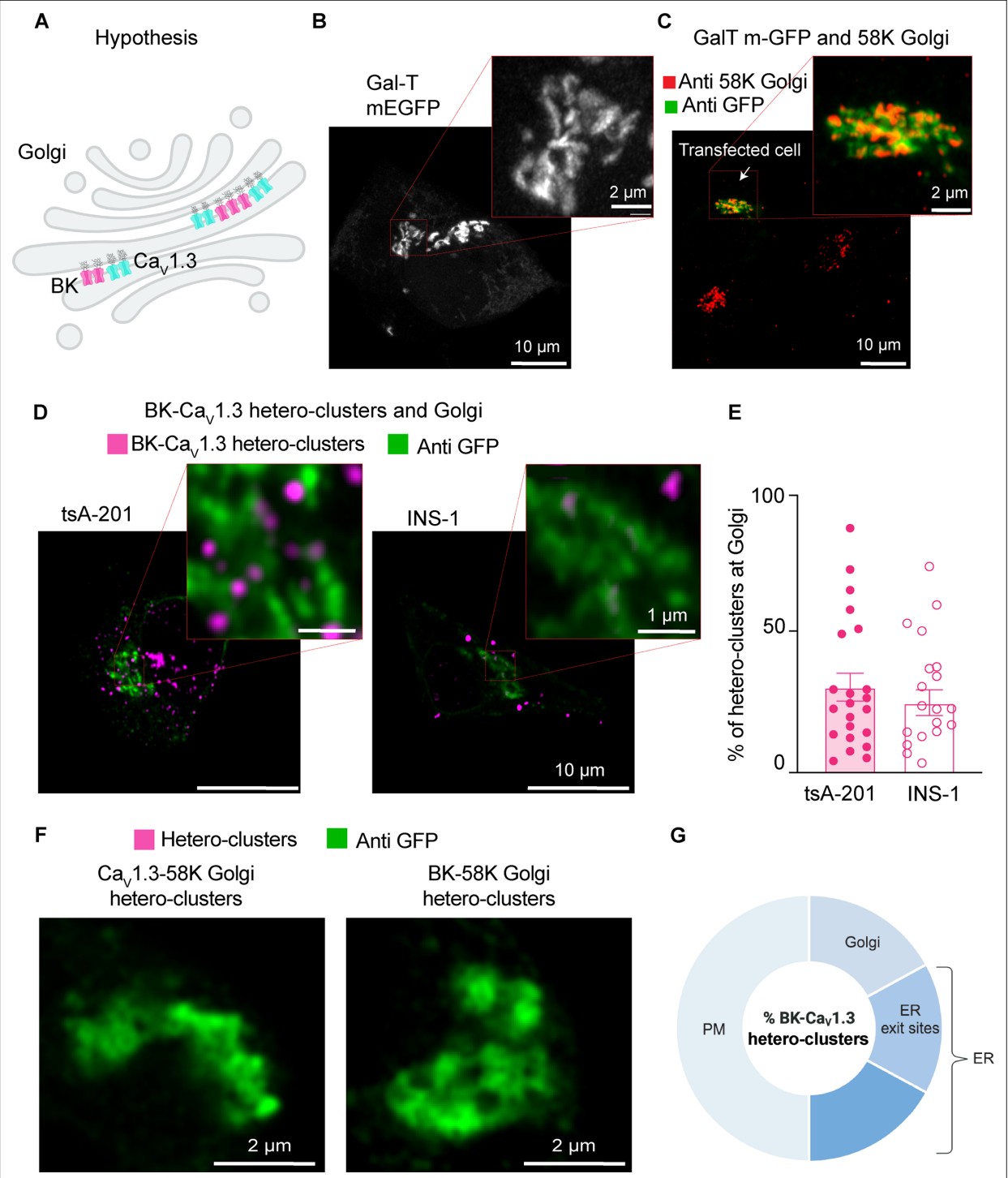

**Figure 4.** BK and Ca$_V$1.3 hetero-clusters go through the Golgi. (**A**) Diagram of the hypothesis: proximity ligation assay (PLA) puncta detecting hetero-clusters between BK (magenta) and Ca$_V$1.3 (cyan) channels can be found at the Golgi membrane. (**B**) Representative image of the Golgi structure with exogenous GFP in INS-1 cells. Cells were transfected with Gal-T-mEGFP. Enlargement is shown in the inset. (**C**) Representative images of fixed cells co-stained with antibodies against Gal-T-mEGFP in green and 58K-Golgi in red. (**D**) Representative images of PLA puncta and Golgi. tsA-201 cells were transfected with BK, Ca$_V$1.3, and Gal-T-mEGFP (left), and INS-1 cells were transfected only with Gal-T-mEGFP (right). PLA puncta are shown in magenta. Golgi is shown in green. (**E**) Scatter dot plot of percentages of BK-Ca$_V$1.3 hetero-clusters found at the Golgi relative to all PLA puncta in tsA-201 and INS1 cells. Data points are from n=22 tsA-201 cells and n=19 INS-1 cells. (**F**) Representative image of PLA puncta and Golgi. tsA-201 cells were transfected with BK, Ca$_V$1.3, and Gal-T-mEGFP. Left: PLA was done against BK and 58K Golgi (magenta), and Golgi is shown in green. Right: PLA was done against Ca$_V$1.3 and 58K Golgi (magenta), and Golgi is shown in green. (**G**) Diagram illustrating our interpretation of percentages of BK-Ca$_V$1.3

*Figure 4 continued on next page*

*Figure 4 continued*

hetero-clusters found in the cell. This illustration is based on results shown in *Figures 1–3* and Figure 4. Percentages were modified to represent overlap of fluorescent signals and limited resolution. We also show that hetero-clusters found in the ER exit sites (ERES) are also accounted in the ER. Scale bars: 10 μm and 2 μm in panels B and C; 10 μm and 1 μm in panel D; 2 μm in panel F.

The online version of this article includes the following figure supplement(s) for figure 4:

**Figure supplement 1.** Ca$_V$1.3 channels do not exhibit proximity interactions with ryanodine receptor type 2 (RyR$_2$).

completely void of fluorescent signal from INS-1 cells probed for bacterial mRNA (negative control, *Figure 5—figure supplement 2A*) and shows images of INS-1 cells probed for mammalian mRNAs expected to be in these cells (positive control, *Figure 5—figure supplement 2B*). We confirmed the specificity of the probes by performing in situ hybridization against *KCNMA1*, *CACNA1D*, *RyR2*, and *SCN9A* in non-transfected cell lines (*Figure 5—figure supplement 2C and D*).

Probes against *GAPDH* mRNA were used as a control in the same cells that we probed for *KCNMA1* and *CACNA1D* using the multiplex capability of this design. Transcripts were detected at different expression levels. *GAPDH* and *CACNA1D* mRNAs were more abundant (134 and 33 mRNA/100 μm², respectively) than *KCNMA1* mRNA (12 mRNA/100 μm², *Figure 5C*). Interestingly, the abundance of *KCNMA1* transcripts correlated more with the abundance of *CACNA1D* transcripts than with the

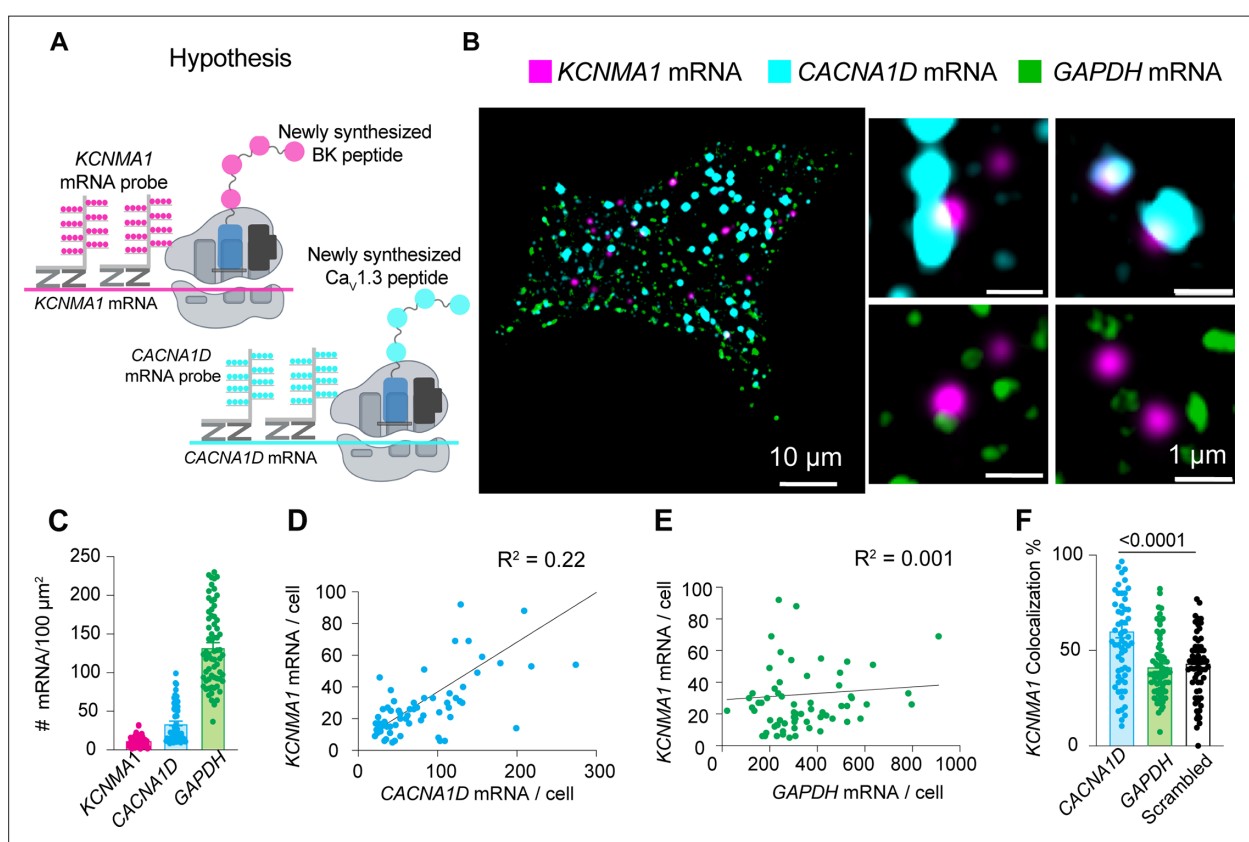

**Figure 5.** BK mRNA (*KCNMA1*) and Ca$_V$1.3 mRNA (*CACNA1D*) colocalize. (**A**) Diagram of the hypothesis: *KCNMA1* and *CACNA1D* mRNAs are found in close proximity to be translated in the same neighborhood. (**B**) Images of fluorescent puncta from RNAscope experiments showing *KCNMA1* mRNA in magenta, *CACNA1D* mRNA in cyan, and *GAPDH* mRNA in green. Right, magnification of three ROIs. (**C**) Comparison of mRNA density of *KCNMA1*, *CACNA1D*, and *GAPDH*. (**D**) Correlation plot of mRNA abundance of *KCNMA1* and *CACNA1D* per cell. (**E**) Correlation plot of mRNA abundance of *KCNMA1* and *GAPDH* per cell. (**F**) Comparison of colocalization between *KCNMA1* mRNA and mRNA from *CACNA1D*, *GAPDH*, and scrambled images of *CACNA1D*. Data points are from n=67 cells. Scale bars are 10 μm and 1 μm in the magnifications. Data were analyzed using ordinary one-way ANOVA with Dunnett's multiple comparisons test.

The online version of this article includes the following figure supplement(s) for figure 5:

**Figure supplement 1.** Illustration of RNAscope methodology.

**Figure supplement 2.** RNA probe validation.

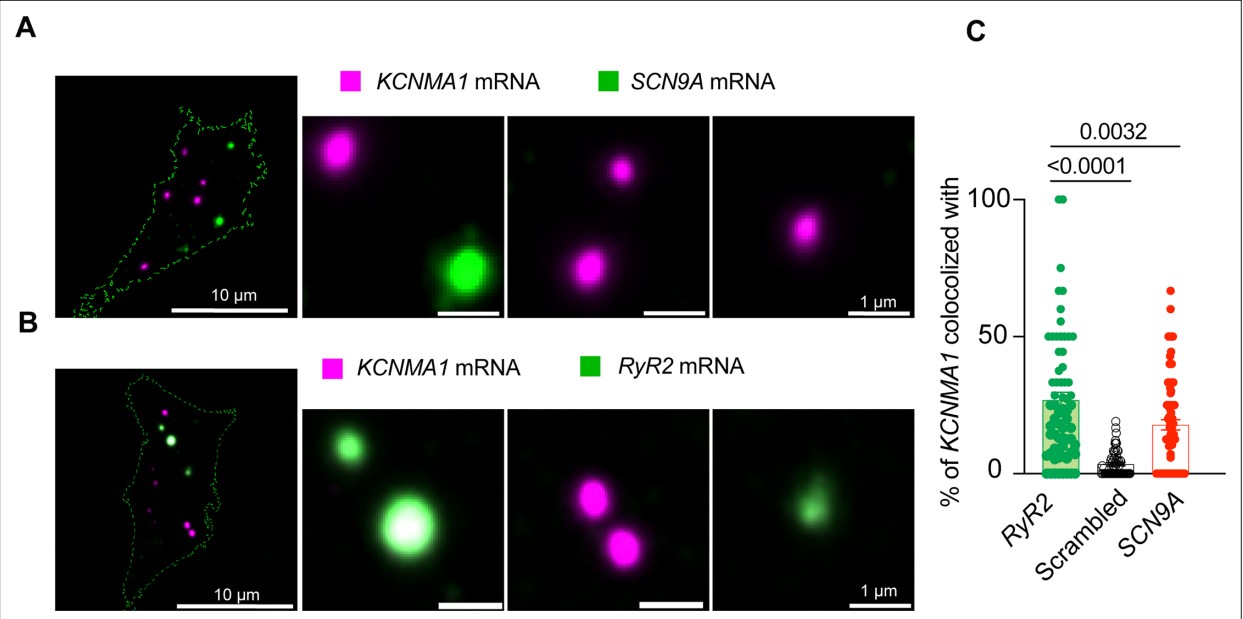

**Figure 6.** BK mRNA (*KCNMA1*) and RyR-2 mRNA (*RyR2*) colocalize. (**A**) Representative confocal images of *KCNMA1* and Na$_V$1.7 (*SCN9A*) mRNA. (**B**) Representative images of *KCNMA1* and *RyR2* mRNA. (**C**) Comparison of the colocalization between *KCNMA1* mRNA and mRNA from *RyR2*, *SCN9A*, and scrambled images of *KCNMA1*. Data points are from n=67 cells. One-way ANOVA. Scale bars are 10 μm and 1 μm in the magnifications. Data were analyzed using ordinary one-way ANOVA with Dunnett's multiple comparisons test.

abundance of *GAPDH*, though with a modest R² value (**Figure 5D and E**). Furthermore, *KCNMA1* and *CACNA1D* mRNA colocalized by 60 ± 4%, which was 20% more than with *GAPDH* mRNA (**Figure 5F**). As an additional control and to rule out the potential influence of differences between *CACNA1D* and *KCNMA*1 abundance, we assessed *CACNA1D* colocalization against randomized, computer-generated *KCNMA1* mRNA signals, where localization was randomized while maintaining the same overall transcript count. The significantly lower (20%) colocalization observed in scrambled conditions compared to genuine BK-Cav1.3 mRNA interactions confirms that proximity is not an artifact of expression levels but reflects a specific spatial association. These results suggest that some fraction of mRNAs for BK and Ca$_V$1.3 channels are translated nearby, so the channel proteins potentially could be inserted into the same regions of the ER. We suggest that the newly synthesized proteins remain together during trafficking through the ER and the Golgi.

Next, we compared the proximity between *KCNMA1* and *SCN9A*. We detected *SCN9A* in the same cells where *KCNMA1* was found (**Figure 6A**). The colocalization between *KCNMA1* and *SCN9A* was only 18 ± 2% (**Figure 6C**), which was less than what was observed with *KCNMA1* and *CACNA1D* (60%, **Figure 5F**). Notably, we observed comparable levels of co-localization between *KCNMA1* and *SCN9A* transcripts in both directions (data not shown), with no statistically significant difference. These findings support the specificity of mRNA co-localization and that *KCNMA1* tends to localize closer to *CACNA1D* than to *SCN9A* or *GAPDH*, supporting the specificity of *KCNMA1* and *CACNA1D* mRNA association.

With the goal of understanding if this concept could apply to other channels, we used the same approach to test a second protein known to provide calcium for BK channel opening. Similar to the coupling between BK and Ca$_V$1.3 channels, RyR2 also can be structurally coupled to BK channels (**Lifshitz et al., 2011**). **Figure 6B** shows high-resolution images of single-molecule in situ hybridization for *RyR2* probed together with *KCNMA1* in INS-1 cells. In the same population of cells, *KCNMA1* and *RyR2* colocalization was 27 ± 3%, which is 1.5 times that with *SCN9A* mRNA. We also assessed *RyR2* colocalization with randomized, computer-generated *KCNMA1* mRNA signals. We found that colocalization between randomized *KCNMA1* and genuine *RyR2* was 87% less colocalized compared to the analysis of genuine *KCNMA1* and genuine *RyR2* (**Figure 6C**), suggesting that *KCNMA1* not only colocalizes with *CACNA1D* but also with mRNA of other known calcium sources.

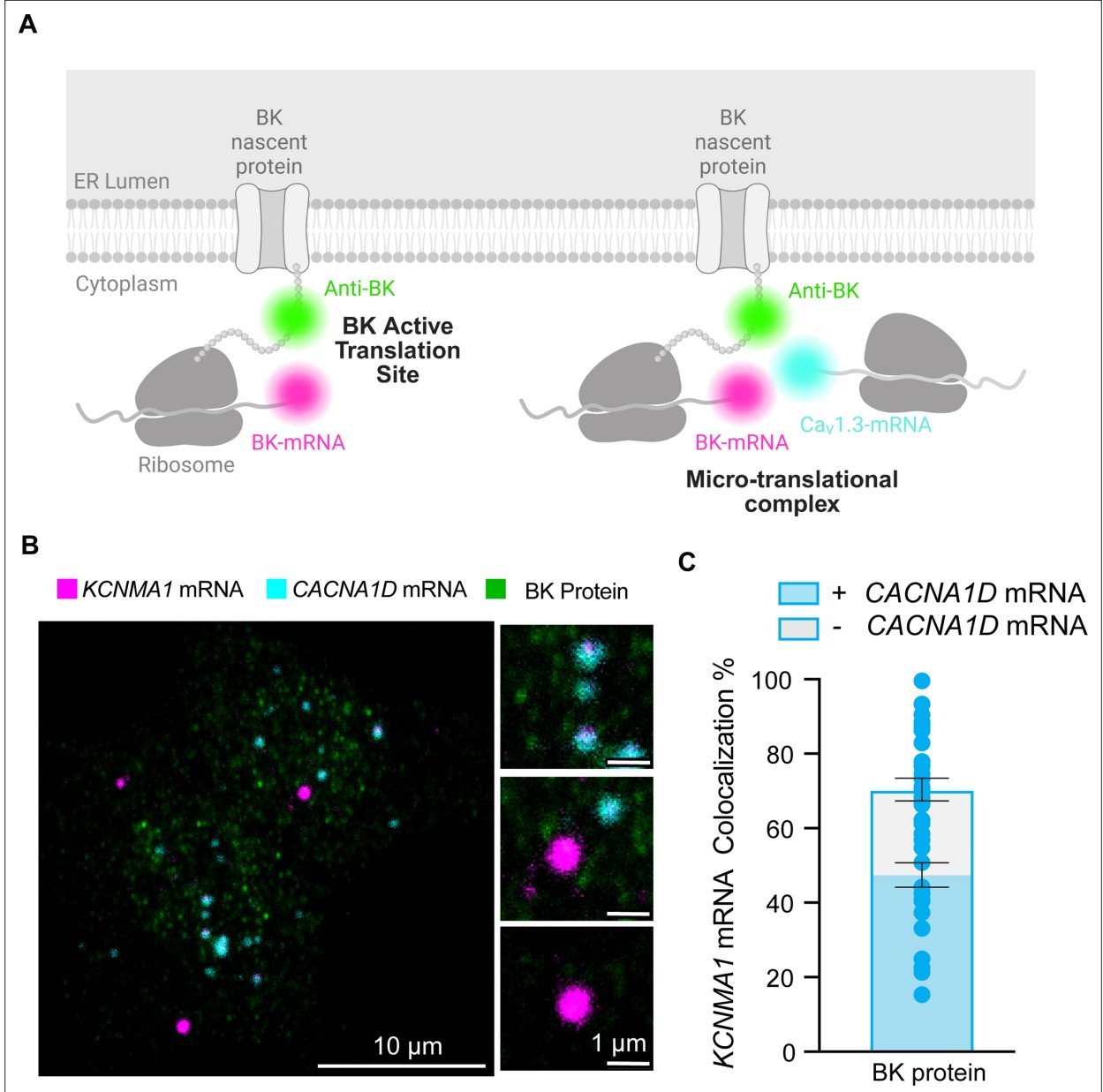

**Figure 7.** BK mRNA (*KCNMA1*) and Ca$_V$1.3 mRNA (*CACNA1D*) colocalize in micro-translational complexes. (**A**) Diagram of the hypothesis: *KCNMA1* mRNAs are found in micro-translational complexes. (**B**) Representative images of *KCNMA1* mRNA in magenta, *CACNA1D* mRNA in cyan, and BK protein in green. (**C**) Comparison of the frequency of colocalization of *KCNMA1* mRNA in active translation and in micro-translational complexes. Data points are from n=57 cells. One-way ANOVA was used as statistical analysis. Scale bars are 10 μm and 1 μm in the magnifications.

To further investigate whether *KCNMA1* and *CACNA1D* are localized in regions of active translation (*Figure 7A*), we performed RNAscope targeting *KCNMA1* and *CACNA1D* alongside immunostaining for BK protein. This strategy enabled us to visualize transcript-protein colocalization in INS-1 cells with subcellular resolution.

By directly evaluating sites of active BK translation, we aimed to determine whether newly synthesized BK protein colocalized with *CACNA1D* mRNA signals (*Figure 7A*). Confocal imaging revealed distinct micro-translational complexes where *KCNMA1* mRNA puncta overlapped with BK protein signals and were located adjacent to *CACNA1D* mRNA (*Figure 7B*). Quantitative analysis showed that 71 ± 3% of all *KCNMA1* colocalized with BK protein signal, which means that they are in active translation. Interestingly, 69 ± 3% of the *KCNMA1* in active translation colocalized with *CACNA1D*

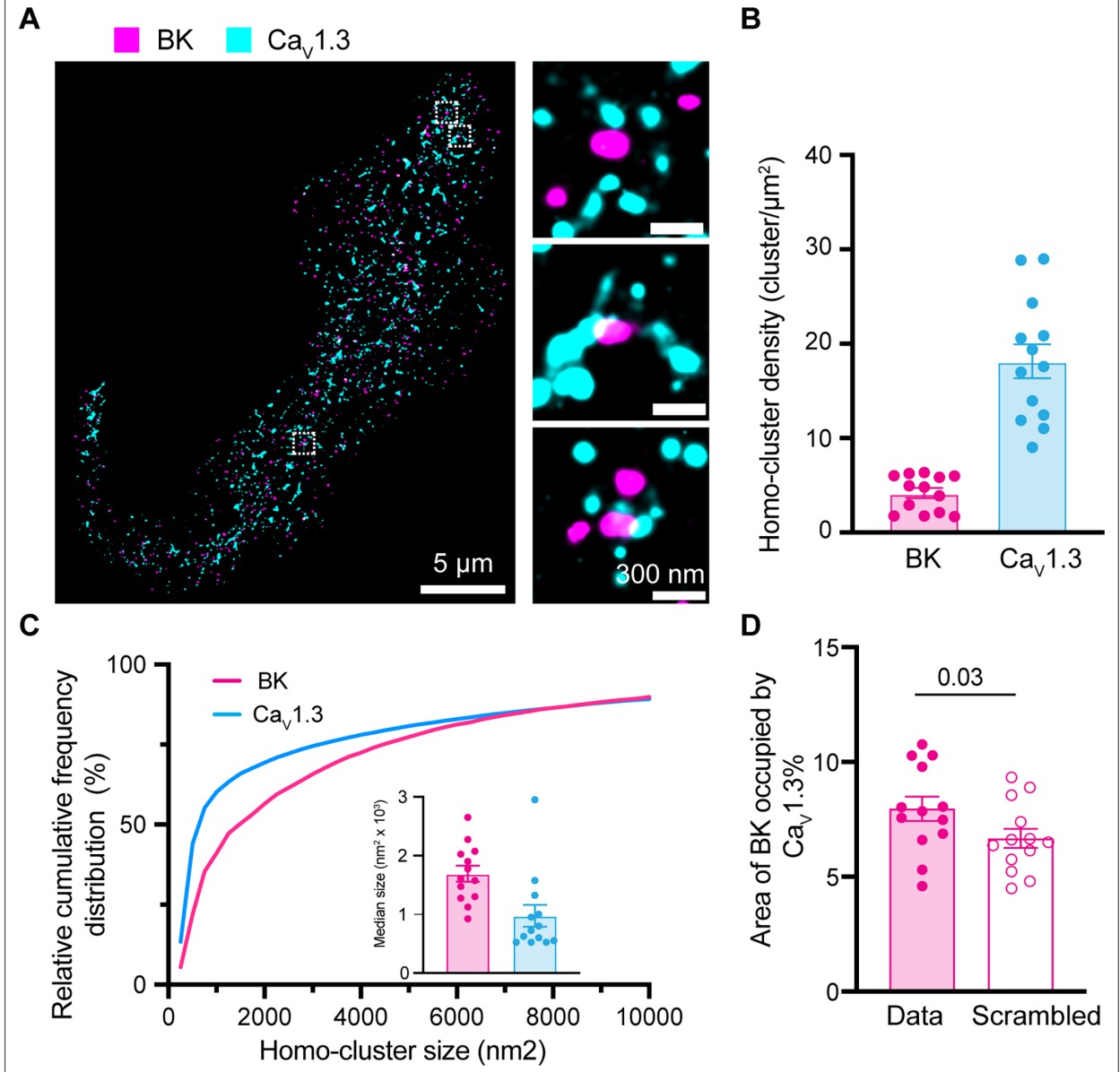

**Figure 8.** Formation of BK and Ca$_V$1.3 hetero-clusters in INS-1 cells. (**A**) Representative localization map of antibodies against BK (magenta) and Ca$_V$1.3 (cyan) channels. Magnifications are shown in the insets on the right. (**B**) Scatter dot plot of homo-cluster densities of BK and Ca$_V$1.3 channels in INS-1 cells. (**C**) Cumulative frequency distributions of homo-cluster sizes of BK and Ca$_V$1.3 channels. Inset compares median size of BK and Ca$_V$1.3 homo-clusters. (**D**) Comparison of colocalization between BK and Ca$_V$1.3 and between scrambled BK and Ca$_V$1.3. Data points are from n=13 cells. Scale bars are 5 µm and 300 nm in the magnifications. Data was analyzed using paired t-tests to evaluate significance.

The online version of this article includes the following figure supplement(s) for figure 8:

**Figure supplement 1.** Organization of BK-Ca$_V$1.3 hetero-clusters in INS-1 cells.

(*Figure 7C*), supporting the existence of functional micro-translational complexes between BK and Ca$_V$1.3 channels.

## BK and Ca$_V$1.3 channels form hetero-clusters at the plasma membrane of INS-1 cells

We previously showed the organization of BK and Ca$_V$1.3 channels in hetero-clusters in tsA-201 cells and neurons, including hippocampal and sympathetic motor neurons (*Vivas et al., 2017*). Our present study utilizes INS-1 cells. We detected single microscopy and determined their degree of

colocalization. *Figure 8A* shows a representative image of the localizations of antibodies against BK and Ca$_V$1.3 channels. Maps were rendered at 5 nm, the pixel size for the images presented. BK-positive pixels formed multi-pixel aggregates, which have been interpreted as homo-clusters of molecules in INS-1 cells. It is important to note that this technique does not allow us to distinguish between labeling of four BK α-subunits within a tetramer and labeling of multiple BK channels in a cluster. Hence, particles smaller than ~1680 nm² may represent either a single tetramer or a cluster. This limitation applies to *Figures 8C and 9D* and does not affect measurements of BK-Ca$_V$1.3 proximity. Ca$_V$1.3 channels also formed homo-clusters. When looking at the formation of BK and Ca$_V$1.3 hetero-clusters, no fixed geometry or stoichiometry was observed. On the contrary, maps showed a distribution of distances between BK and Ca$_V$1.3 detected pixels. In some cases, BK and Ca$_V$1.3 channels were perfectly overlapping, as if the detection of the antibodies could not be separated by the 20 nm resolution. In other cases, BK and Ca$_V$1.3 channels were adjacent (less than 5 nm), and in other cases, BK homo-clusters were surrounded by a variable number of Ca$_V$1.3 channels.

To provide a more quantitative description of the maps, we measured cluster density, cluster size, and colocalization. We used the particle analysis tool of the software ImageJ for these measurements. In this analysis, a particle represents positive pixels irrespective of the number of pixels composed. INS-1 cells showed four times as many Ca$_V$1.3 particles as BK particles (*Figure 8B*). *Figure 8C* shows the frequency distribution of particle size, where the median area is 1691 nm² for BK and 975 nm² for Ca$_V$1.3. The colocalization between BK and Ca$_V$1.3 clusters (*Figure 8D*) was 7.5%, a value higher than that obtained from scrambled image controls. Notably, 37±3% BK channels are at 200 nm or less from one or more Ca$_V$1.3, while 15 ± 2% of Ca$_V$1.3 channels are at 200 nm or less from one or more BK (*Figure 8—figure supplement 1A*). Furthermore, the distribution of the nearest distance between BK and Ca$_V$1.3 in INS-1 cells (*Figure 8—figure supplement 1B*, magenta) shows a large proportion of BK channels (around 40%) at <50 nm from any Ca$_V$1.3 channels. Together, these results suggest that INS-1 cells also exhibit nanodomains containing BK-Ca$_V$1.3 hetero-clusters at the plasma membrane.

## Hetero-clusters of BK and Ca$_V$1.3 channels are detected at the plasma membrane soon after their expression

In light of our results, our current model is that BK and Ca$_V$1.3 hetero-clusters form prior to their insertion into the plasma membrane. One prediction based on this model is that channels inserted in the plasma membrane would appear as hetero-clusters already at early time points after the start of their synthesis. To test this prediction, we transfected tsA-201 cells with BK and Ca$_V$1.3 channels and performed a chase experiment to detect their presence at the plasma membrane using super-resolution microscopy. We measured particle size, density, and colocalization at 18, 24, and 48 hr after transfection. Notably, although the channels were transfected simultaneously, BK particles were detected in the plasma membrane 18 hr after transfection, whereas Ca$_V$1.3 particles were detected 24 hr after transfection (*Figure 9A–C*). The distribution of BK and Ca$_V$1.3 particle size did not change with the time after transfection (*Figure 9D and E*), in agreement with *Sato et al., 2019*.

In support of our prediction, we found particles of Ca$_V$1.3 near BK particles as soon as the expression of Ca$_V$1.3 channels was detected in the plasma membrane (24 hr after transfection, enlarged *Figure 9B*). To test this hypothesis further, we compared plots of localization preference at 24 and 48 hr. These plots were constructed by measuring the percentage of Ca$_V$1.3 area occupying concentric regions of 20 nm width around BK particles. Values were normalized to the percentage of colocalization found at 200 nm from BK. *Figure 9F* shows that the localization preference plots are identical at 24 and 48 hr, consistent with the hypothesis that both channels are inserted together into the plasma membrane already as hetero-clusters.

## Discussion

### Intracellular assembly of BK and Ca$_V$1.3 channels

Previous work from our group has revealed the organization of hetero-clusters of large-conductance calcium-activated potassium (BK) channels and voltage-gated calcium channels, particularly Ca$_V$1.3, within nanodomains (*Vivas et al., 2017*). This spatial organization is crucial for the functional coupling that enables BK channels to respond promptly to calcium influx, thereby modulating cellular excitability. While physical details of BK-Ca$_V$1.3 interactions are not fully elucidated, the evidence suggests

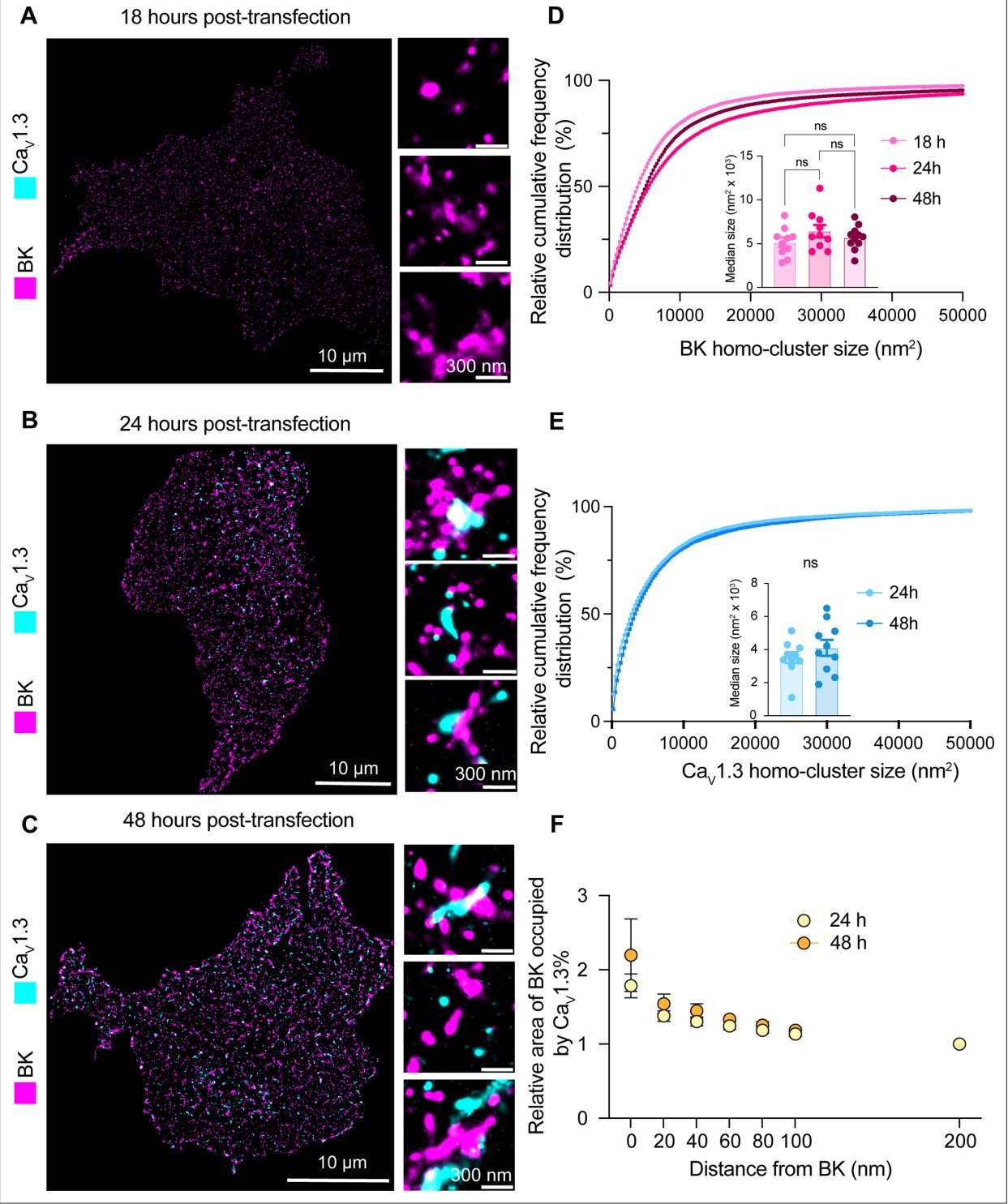

**Figure 9.** Hetero-clusters of BK and Ca$_V$1.3 channels are detected at the plasma membrane soon after their expression begins. (**A–C**) Representative localization maps of antibodies against BK and Ca$_V$1.3 channels. (**A**) 18 hr, (**B**) 24 hr, or (**C**) 48 hr after DNA transfection into cells. Enlargements are shown in insets. (**D**) Cumulative frequency distributions of BK homo-cluster size at 18, 24, and 48 hr. Inset compares median BK homo-cluster areas. (**E**) Cumulative frequency distributions of Ca$_V$1.3 homo-clusters at 24 and 48 hr. The inset compares median Ca$_V$1.3 cluster areas. Ca$_V$1.3 clusters are not present at the 18 hr time point. (**F**) Comparison of colocalization plots between BK and Ca$_V$1.3 channels at 24 and 48 hr time points. Data points are from n=10 cells. Scale bars are 10 µm and 300 nm in enlargements. Statistical significance was assessed using ordinary one-way ANOVA followed by Dunnett's multiple comparisons test and unpaired t-test in panels D and E, respectively.

that BK channels interact with both the Ca$_V$ α1-subunit and its auxiliary subunits (*Rehak et al., 2013*; *Zhang et al., 2018*).

Our findings highlight the intracellular assembly of BK-Ca$_V$1.3 hetero-clusters, though limitations in resolution and organelle-specific analysis prevent precise quantification of the proportion of intracellular complexes that ultimately persist on the cell surface. While our data confirms that hetero-clusters form before reaching the plasma membrane, it remains unclear whether all intracellular hetero-clusters transition intact to the membrane or undergo rearrangement or disassembly upon insertion. Future studies utilizing live-cell tracking and high-resolution imaging will be valuable in elucidating the fate and stability of these complexes after membrane insertion.

A stochastic model of ion channel homo-cluster formation in the plasma membrane proposes that random yet probabilistic interactions between ion channels contribute to their formation (*Sato et al., 2019*). These clusters are stabilized and organized within specific membrane regions through biophysical mechanisms such as membrane curvature. Recent findings on spatial organization of G-protein-coupled receptors provide additional support for this framework, demonstrating that the coupling of membrane proteins to the curvature of the plasma membrane acts as a driving force for their clustering into functional domains (*Kockelkoren et al., 2024*). Yet, the mechanisms regulating the spatial organization of hetero-clusters remain less known.

## Colocalization and trafficking dynamics

The colocalization of BK and Ca$_V$1.3 channels in the ER and at ER exit sites before reaching the Golgi suggests a coordinated trafficking mechanism that facilitates the formation of multichannel complexes. This functional coupling is crucial for calcium signaling and membrane excitability (*Berkefeld et al., 2010*; *Loane et al., 2007*). Given the distinct roles of these compartments, colocalization at the ER and ER exit sites may reflect transient proximity rather than stable interactions. Their presence in the Golgi further suggests that posttranslational modifications and additional assembly steps occur before plasma membrane transport, providing further insight into hetero-cluster maturation and sorting events. By examining BK-Ca$_V$1.3 hetero-cluster distribution across these trafficking compartments, we ensure that observed colocalization patterns are considered within a broader framework of intracellular transport mechanisms (*Boncompain and Perez, 2013*). Previous studies indicate that ER exit sites exhibit variability in cargo retention and sorting efficiency (*Kurokawa and Nakano, 2019*), emphasizing the need for careful evaluation of colocalization data. Accounting for these complexities allows for a robust assessment of signaling complexes formation and trafficking pathways.

## BK surface expression and independent trafficking pathways

BK surface expression in the absence of Ca$_V$1.3 indicates that its trafficking does not strictly rely on Ca$_V$1.3-mediated interactions. Since BK channels can be activated by multiple calcium sources, their presence in intracellular compartments suggests that their surface expression is governed by intrinsic trafficking mechanisms rather than direct calcium-dependent regulation. While some BK and Ca$_V$1.3 hetero-clusters assemble into signaling complexes intracellularly, other BK channels follow independent trafficking pathways, demonstrating that complex formation is not obligatory for all BK channels. Differences in their transport kinetics further reinforce the idea that their intracellular trafficking is regulated through distinct mechanisms. Studies have shown that BK channels can traffic independently of Ca$_V$1.3, relying on alternative calcium sources for activation (*Shah et al., 2021*; *Chen et al., 2023c*). Additionally, Ca$_V$1.3 exhibits slower synthesis and trafficking kinetics than BK, emphasizing that their intracellular transport may not always be coordinated. These findings suggest that BK and Ca$_V$1.3 exhibit both independent and coordinated trafficking behaviors, influencing their spatial organization and functional interactions. Future experiments using inducible constructs to precisely control transcription timing will enable more precise quantification of hetero-cluster formation in the ER compartment prior to plasma membrane insertion and reduce the variability introduced by differences in expression timing after plasmid transfection.

## mRNA colocalization and protein trafficking

The colocalization of mRNAs encoding BK and Ca$_V$1.3 channels suggests a coordinated translation mechanism that facilitates their proximal synthesis and subsequent assembly into functional complexes. As a mechanism, mRNA localization enhances protein enrichment at functional sites,

coordinating with translational control to position ion channels at specific subcellular domains; ion channel positioning is especially critical for neurons and cardiomyocytes (*Vacher et al., 2008*; *Medioni et al., 2012*). By synthesizing these channels in close proximity, channels can be efficiently assembled into functional units, making this process crucial for precise targeting and trafficking of ion channels to specific subcellular domains, thereby ensuring proper cellular function and efficient signal transduction (*Blandin et al., 2021*; *Dai, 2023*; *Wang et al., 2023*). Disruption in mRNA localization or protein trafficking can lead to ion channel mislocalization, resulting in altered cellular function and disease (*Wang et al., 2016*; *Jameson et al., 2023*). In cardiac cells, precise trafficking of ion channels to specific membrane subdomains is crucial for maintaining normal electrical and mechanical functions, and its disruption contributes to heart disease and arrhythmias (*Anderson et al., 2006*; *Smyth and Shaw, 2010*; *Basheer and Shaw, 2016*). Similarly, in neurons, the localization and translation of mRNA at synaptic sites are essential for synaptic plasticity, and disturbances can lead to neurological disorders (*Eliscovich et al., 2017*; *Yoon et al., 2016*).

## Co-translational regulation and functional coordination

The potential co-translational association of transcripts encoding BK and $Ca_V1.3$ ion channels, as well as other known calcium sources like RyR2, may serve as a mechanism to ensure the precise stoichiometry and assembly of functional channel complexes. It is important to note that while our data suggest mRNA coordination, additional experiments are required to directly assess co-translation. In parallel, the spatial organization of BK channels with other calcium sources such as RyR2—particularly in airway myocytes—has been shown to facilitate efficient BK activation by localized $Ca^{2+}$ sparks (*Rehak et al., 2013*). Co-translational association of transcripts encoding $K_V1.3$ channels was the first example in which the interaction of nascent $K_V1.3$ N-termini facilitates proper tertiary and quaternary structure required for oligomerization (*Robinson and Deutsch, 2005*; *Tu and Deutsch, 1999*). Similarly, co-translational heteromeric association of hERG1a and hERG1b subunits ensures that cardiac $IK_r$ currents exhibit the appropriate biophysical properties and magnitude necessary for normal ventricular action potential shaping (*Liu et al., 2016*). The precise mechanism by which BK transcripts are associated with the transcripts of calcium sources like $Ca_V1.3$ and RyR2 channels remains to be elucidated. While it is conceivable that complementary base pairing or tertiary structural interactions play a role, RNA-binding proteins (RBPs) are likely key mediators of these associations.

## Conclusion and future directions

This study provides novel insights into the organization of BK and $Ca_V1.3$ channels in hetero-clusters, emphasizing their assembly within the ER, at ER exit sites, and within the Golgi. Our findings suggest that BK and $Ca_V1.3$ channels begin assembling intracellularly before reaching the plasma membrane, shaping their spatial organization and potentially facilitating functional coupling. While this suggests a coordinated process that may contribute to functional coupling, further investigation is needed to determine the extent to which these hetero-clusters persist upon membrane insertion. While our study advances the understanding of BK and $Ca_V1.3$ hetero-cluster assembly, several key questions remain unanswered. What molecular machinery drives this colocalization at the mRNA and protein level? How do disruptions to complex assembly contribute to channelopathies and related diseases? Additionally, a deeper investigation into the role of RBPs in facilitating transcript association and localized translation is warranted.

# Materials and methods

**Key resources table**

| Reagent type (species) or resource | Designation | Source or reference | Identifiers | Additional information |
| --- | --- | --- | --- | --- |
| Cell line (human) | tsA-201 | Sigma | RRID:CVCL_2737 | Authentication: STR profiling; Mycoplasma: negative (see Materials and methods) |
| Cell line (rat) | INS-1 | Sigma | RRID:CVCL_0352 | Authentication: STR profiling/DNA barcoding; Mycoplasma: negative (see Materials and methods) |

*Continued on next page*

*Continued*

| Reagent type (species) or resource | Designation | Source or reference | Identifiers | Additional information |
|---|---|---|---|---|
| Recombinant DNA reagent | CaV1.3 | Addgene | RRID:Addgene_49333; Addgene:49333 | Plasmid expressing CACNA1D (rat) |
| Recombinant DNA reagent | Slo1 (BK) | Addgene | RRID:Addgene_113566; Addgene:113566 | Mouse BK channel α-subunit expression vector |
| Recombinant DNA reagent | KDEL-mox-GFP (ERmoxGFP) | Addgene | RRID:Addgene_68072; Addgene:68072 | ER-localized moxGFP with KDEL retention sequence |
| Recombinant DNA reagent | pmGFP-Sec16 | Addgene | RRID:Addgene_15775; Addgene:15775 | pmGFP-Sec16S mammalian expression plasmid |
| Recombinant DNA reagent | Golgi-mEGFP | Addgene | RRID:Addgene_182877; Addgene:182877 | Golgi-targeted mEGFP |
| Recombinant DNA reagent | pPH-PLC-δ-GFP (PH-PLCD1-GFP) | Addgene | RRID:Addgene_51407; Addgene:51407 | Biosensor for PI(4,5)P2 |
| Recombinant DNA reagent | CaVβ3 | Diane Lipscombe, Brown University | other | Auxiliary subunit for $Ca_V1.3$; repository ID not available; contact depositor |
| Recombinant DNA reagent | CaVα2δ1 | Diane Lipscombe, Brown University | Other | Auxiliary subunit for $Ca_V1.3$; repository ID not available; contact depositor |
| Sequence-based reagent (rat) | RNAscope 3-plex Positive Control Probes | Advanced Cell Diagnostics | Cat. No. 320871 | Species-specific housekeeping targets for RNAscope; 3-plex positive control |
| Sequence-based reagent (rat) | RNAscope 3-plex Negative Control Probe (dapB) | Advanced Cell Diagnostics | Cat. No. 320871 | Negative control probe targeting bacterial dapB |
| Sequence-based reagent (rat) | RNAscope Probe—Rn-Ryr | Advanced Cell Diagnostics | Cat. No. 2560931 | Custom target probe |
| Sequence-based reagent (rat) | RNAscope Probe—Rn-Scn9a | Advanced Cell Diagnostics | Cat. No. 317851 | Custom target probe |
| Sequence-based reagent (rat) | RNAscope Probe—Rn-Kcnma1-C3 | Advanced Cell Diagnostics | Cat. No. 1108261-C3 | Channel C3 probe |
| Sequence-based reagent (rat) | RNAscope Probe—Rn-Cacna1d-C2 | Advanced Cell Diagnostics | Cat. No. 409361-C2 | Channel C2 probe |
| Sequence-based reagent (rat) | RNAscope Probe—Rn-Gapdh | Advanced Cell Diagnostics | Cat. No. 409821 | Housekeeping gene control |
| Antibody | Anti-CaV1.3 (Rabbit polyclonal) | Drs. William Catterall and Ruth Westenbroek | | Rabbit polyclonal primary recognizing residues 809–825 (II–III loop), (1:100), (1 µl) |
| Antibody | Anti-Slo1 (clone L6/60) (Mouse monoclonal) | Millipore Sigma | RRID:AB_10805948; Cat. No. MABN70 | Mouse monoclonal, (1:100), (1 µl) |
| Antibody | Anti-GFP (Goat polyclonal) | Abcam | RRID:AB_305643; Cat. No. ab6673 | Rabbit polyclonal to full-length Aequorea victoria GFP, (1:100), (1 µl) |
| Antibody | Anti-58K Golgi (Mouse monoclonal) | Abcam | RRID:AB_2107005; Cat. No. ab27043 | Mouse monoclonal recognizing Golgi protein (1:100), (1 µl) |
| Antibody | Donkey anti-mouse Alexa 647 (Donkey polyclonal) | Invitrogen (Molecular Probes) | RRID:AB_2536183; Cat. No. A-31573 | Secondary antibody donkey polyclonal, (1:1000), (1 µl) |
| Antibody | Donkey anti-rabbit Alexa 555 (Donkey polyclonal) | Invitrogen (Molecular Probes) | RRID:AB_2534017; Cat. No. A10042 | Secondary antibody, donkey polyclonal, (1:1000), (1 µl) |
| Antibody | Donkey anti-goat Alexa 488 (Donkey polyclonal) | Invitrogen (Molecular Probes) | RRID:AB_162542; Cat. No. A-31571 | Secondary antibody, donkey polyclonal, (1:1000), (1 µl) |
| Antibody | Duolink In Situ PLA probe anti-rabbit PLUS (Donkey polyclonal) | Sigma | RRID:AB_2810940 | Proximity ligation assay probe, donkey polyclonal, (1:20), (2 µl) |

*Continued*

| Reagent type (species) or resource | Designation | Source or reference | Identifiers | Additional information |
|---|---|---|---|---|
| Antibody | Duolink In Situ PLA probe anti-mouse PLUS (Donkey polyclonal) | Sigma | RRID:AB_2810939 | Proximity ligation assay probe, donkey polyclonal, (1:20), (2 µl) |
| Commercial assay or kit | Duolink In Situ Red Starter Kit | Sigma | Cat. No. DUO92008 | PLA detection kit |
| Chemical compound, drug | Lipofectamine 3000 | Invitrogen | Cat. No. L3000001/008/015 (series) | Transfection reagent; see Thermo Fisher catalog for sizes |
| Chemical compound, drug | ProLong Gold Antifade Mountant with DAPI | Invitrogen | Cat. No. P36931 | Mounting medium with DAPI |
| Commercial assay or kit | RNAscope Multiplex Fluorescent v2 Assay | Advanced Cell Diagnostics (Bio-Techne) | Cat. No. 323270 (with TSA Vivid dyes)/323100 (reagent kit v2) | Manual multiplex fluorescent RNA ISH assay |
| Software, algorithm | Prism | GraphPad | RRID:SCR_002798 | Version 10 |
| Software, algorithm | Excel | Microsoft | RRID:SCR_016137 | Microsoft 365 build |
| Software, algorithm | ImageJ | NIH | RRID:SCR_003070 | ImageJ2/Fiji build |
| Software, algorithm | NImOS | ONI | | Version: v1.18.3 |

## Cell culture

We used tsA-201 cells to co-express BK and Ca$_V$1.3 channels heterologously. Cells were grown in DMEM (Gibco) supplemented with 10% fetal bovine serum and 0.2% penicillin/streptomycin. We used rat insulinoma (INS-1) cells to study endogenous levels of transcripts and proteins of channels. INS-1 cells were cultured in RPMI high glutamate medium (Gibco) with 10% fetal bovine serum, 0.2% penicillin/streptomycin, 10 mM HEPES (Gibco), 1 mM sodium pyruvate (Gibco), and 50 µM 2-mercaptoethanol. Both cell types were passaged twice a week and incubated in 5% CO$_2$ at 37°C. Cell lines tsA-201 (human) and INS-1 (rat) were purchased from Sigma. According to the manufacturer, these cell lines were authenticated and tested for mycoplasma contamination prior to shipment. No additional authentication or contamination testing was performed by the authors.

## Plasmids and transfection

Cells were transfected with 0.1–0.4 µg DNA per plasmid and plated for 24 hr on poly-D-lysine-coated coverslips. Lipofectamine 3000 (Invitrogen, RRID:L30000) was used for the transfection. DNA clones of Ca$_V$1.3, BK channels, PH-PLCδ GFP, ER moxGFP, pmGFP-Sec16S, and Golgi-mGFP were obtained from Addgene (RRID:SCR_002037). The Ca$_V$1.3 α-subunit construct used in our study corresponds to the rat Ca$_V$1.3e splice variant containing exons 8a, 11, 31b, and 42a, with a deletion of exon 32. The BK channel construct corresponds to the VYR splice variant of the mouse BKα-subunit (KCNMA1). Auxiliary subunits for Ca$_V$1.3 channels, Ca$_V$β3 and Ca$_V$α2δ1 (from Diane Lipscombe, Brown University, RI, USA), were transfected as well. No BK channel auxiliary subunits were transfected.

## Antibodies

Ca$_V$1.3 channels were immuno-detected with a rabbit primary antibody recognizing residues 809–825 located at the intracellular II-III loop of the channel (DNKVTIDDYQEEAEDKD), kindly provided by Drs. William Catterall and Ruth Westenbroek (*Hell et al., 1993*). BK channels were detected using the anti-Slo1 mouse monoclonal antibody clone L6/60. The goat polyclonal GFP antibody was against the recombinant full-length protein corresponding to *Aequorea victoria* GFP. Anti-58K Golgi protein antibody was used to mark the Golgi. Specificity of antibodies was tested in untransfected tsA-201 cells (*Figure 2—figure supplement 1*). The secondary antibodies tagged with Alexa Fluor dyes were

donkey anti-mouse Alexa Fluor 647, donkey anti-rabbit Alexa Fluor 555, donkey anti-goat Alexa Fluor 488 (Molecular Probes).

## Immunostaining

Cells were fixed with freshly prepared 4% paraformaldehyde for 10 min. After washing, aldehydes were reduced with 0.1% NaBH$_4$ for 5 min and then washed again. Nonspecific binding was blocked with 3% bovine serum albumin (Thermo Scientific). Cells were permeabilized with 0.25% vol/vol Triton X-100 in PBS for 1 hr. Primary antibodies were used at 10 µg/ml in blocking solution and incubated overnight at 4°C. After washing, secondary antibodies at 2 µg/ml were incubated for 1 hr at 21°C. Washing steps indicated in all methods include 3 cycles of rinsing and rocking for 5 min with PBS at 21°C. Cells were imaged using an inverted AiryScan microscope or an ONI Nanoimager with super-resolution capabilities, in total internal reflection fluorescence mode, and with a Z-resolution of 50 nm.

## Proximity ligation assay

Cells were fixed with freshly prepared 4% paraformaldehyde for 10 min. After washing, aldehydes were reduced with 50 mM glycine for 15 min. After another round of washes, PLA was performed according to the manufacturer's instructions (Duolink In Situ Red Starter Kit). Cells were blocked and permeabilized with Duolink blocking solution. Primary antibodies were used at 10 µg/ml in Duolink antibody diluent and incubated overnight at 4°C. The Duolink In Situ PLA probe anti-rabbit PLUS and anti-mouse MINUS were used as secondary antibodies, followed by ligation and amplification. For PLA combined with immunostaining, PLA was followed by a secondary antibody incubation with Alexa Fluor 488 at 2 µg/ml for 1 hr at 21°C. Since GFP fluorescence fades significantly during the PLA protocol, resulting in reduced signal intensity and poor image resolution, GFP was labeled using an antibody rather than relying on its intrinsic fluorescence. Coverslips were mounted using ProLong Gold Antifade Mountant with DAPI. Cells were imaged using an inverted Zeiss AiryScan microscope.

## Single-molecule fluorescence in situ hybridization (RNAscope)

Manual RNAscope assay was performed using RNAscope Multiplex Fluorescent V2 Assay according to the manufacturer's protocol. The RNAscope assay consists of target probes and a signal amplification system composed of a preamplifier, amplifier, and label probe. A schematic RNAscope assay procedure is shown in *Figure 5—figure supplement 1*. The probes against the mRNAs of interest and tested in this work were designed by Advanced Cell Diagnostics. Briefly, cells were fixed with 4% paraformaldehyde for 30 min, washed, dehydrated, and then rehydrated with ethanol, and permeabilized with 0.1% Tween-20 in PBS. Next, cells were quenched with H$_2$O$_2$ and treated with Protease III. Probes were hybridized for 2 hr at 40°C followed by RNAscope amplification and then fluorescence detection. Coverslips were mounted using ProLong Gold Antifade Mountant with DAPI. We used the following RNAscope probes: RNAscope 3-plex Positive Control Probes, RNAscope 3-plex negative control probes, RNAscope Probe-Rn-Ryr, RNAscope Probe-Rn-Scn9a, RNAscope Probe-Rn-Kcnma1-C3, RNAscope Probe-Rn-Cacna1d-C2, and RNAscope Probe-Rn-Gapdh. Cells were imaged on the inverted AiryScan microscope. For PLA and RNAscope experiments, we used custom-made macros written in ImageJ. Processing of PLA data included background subtraction. To assess colocalization, fluorescent signals were converted into binary images, and channels were multiplied to identify spatial overlap. Specificity of RNAscope probes was tested in untransfected tsA-201 cells (*Figure 5—figure supplement 2*). For RNAscope combined with immunostaining, RNAscope was followed by blocking in PBS supplemented with 0.01% Tween-20 and 3% BSA for 1 hr at 21°C. Samples were then probed for BK protein using primary antibody overnight at 4°C followed by secondary antibody incubation with Alexa Fluor 488 at 2 µg/ml for 1 hr at 21°C. Coverslips were mounted using ProLong Gold Antifade Mountant. Cells were imaged using an inverted Zeiss AiryScan microscope.

## High-resolution imaging

Cells were imaged using an inverted AiryScan microscope (Zeiss LSM 880) run by ZEN black v2.3 software and equipped with a plan apochromat 63× oil immersion objective with 1.4 NA. Fluorescent dyes were excited with a 405 nm diode, 458–514 nm argon, 561 nm, or 633 nm laser. Emission light was detected using an Airyscan 32 GaAsP detector and appropriate emission filter sets. The point

spread functions were calculated using ZEN black software and 0.1 µm fluorescent microspheres. The temperature inside the microscope housing was 22°C. Images were analyzed using ImageJ (NIH).

### Super-resolution imaging

Direct stochastic optical reconstruction microscopy (dSTORM) images of BK and $Ca_V1.3$ overexpressed in tsA-201 cells were acquired using an ONI Nanoimager microscope equipped with a 100× oil immersion objective (1.4 NA), an XYZ closed-loop piezo 736 stage, and triple emission channels split at 488, 555, and 640 nm. Samples were imaged at 35°C. For single-molecule localization microscopy, fixed and stained cells were imaged in GLOX imaging buffer containing 10 mM β-mercaptoethylamine, 0.56 mg/ml glucose oxidase, 34 µg/ml catalase, and 10% wt/vol glucose in Tris-HCl buffer. Single-molecule localizations were filtered using NImOS software (v.1.18.3, ONI). Localization maps were exported as TIFF images with a pixel size of 5 nm. Maps were further processed in ImageJ (NIH) by thresholding and binarization to isolate labeled structures. To assess colocalization between the signal from two proteins, binary images were multiplied. Particles smaller than 400 $nm^2$ were excluded from the analysis to reflect the spatial resolution limit of STORM imaging (20 nm) and the average size of BK channels. To examine spatial localization preference, binary images of BK were progressively dilated to 20 nm, 40 nm, 60 nm, 80 nm, 100 nm, and 200 nm to expand their spatial representation. These modified images were then multiplied with the $Ca_V1.3$ channel to quantify colocalization and determine BK occupancy at increasing distances from $Ca_V1.3$. To ensure consistent comparisons across distance thresholds, data were normalized using the 200 nm measurement as the highest reference value, set to 1.

### Image scrambling

Images were binarized as TIFF images, and their respective cell perimeter coordinates were exported as CSV files by ImageJ. Processed binary images were then analyzed by our SpotScrambler (https://github.com/jehuang2/SpotScrambler, copy archived at *Huang, 2025*) Python program. SpotScrambler first extracts the areas of fluorescent particles in the binary image. SpotScrambler then redraws the fluorescent particles as circles at randomized coordinates within cell perimeter boundaries. SpotScrambler accurately preserves particle number and particle sizes, averaging less than 1% difference in total area of fluorescent particles between pre-SpotScrambler and post-SpotScrambler images. To ensure reliable randomization for each experiment, results were averaged between three trials of SpotScrambler.

### Data analysis

Excel (Microsoft) and Prism (GraphPad) were used to analyze data. ImageJ was used to process images. One-way ANOVA and non-parametric statistical test (Mann-Whitney-Wilcoxon) were used to test for statistical significance. p-Values<0.05 were deemed statistically significant. The number of cells used for each experiment is detailed in each figure legend.

### Materials availability statement

All newly created materials used in this study are available for public access. Plasmid constructs have been deposited in Addgene (repository links and RRIDs are provided in the Materials and methods section). RNA probes utilized for in situ hybridization are proprietary to ACD Biotechnology and are not accessible for redistribution. Complete details regarding materials, including catalog numbers and RRIDs, are provided in the Materials and methods section and in the Supporting Data Excel sheet.

All datasets supporting the findings of this study have been deposited in Dryad and are publicly available (https://doi.org/10.5061/dryad.63xsj3vfq).

## Acknowledgements

We thank Gail Robertson for her immense support and influence in designing and conducting this project. We thank Alexey Merz for his support and feedback through this project. We thank Bertil Hille for his contribution to editing the writing of this project. We thank Martina Hunt, Maria Elena Danoviz, Paula Martinez-Feduchi, Wendy Piñon-Teal, and Raul Riquelme for reading and providing feedback on the manuscript. We thank William Catterall and Ruth E Westenbroek for providing the

antibody against Ca$_v$1.3. This study was supported by the National Institutes of Health MIRA R35 GM142690 to OV CM was supported by HL162609 and the Freeman Hrabowski HHMI Scholars program.

## Additional information

### Funding

| Funder | Grant reference number | Author |
| --- | --- | --- |
| National Institute of General Medical Sciences | R35GM142690 | Oscar Vivas |
| National Heart Lung and Blood Institute | HL162609 | Claudia M Moreno |
| Howard Hughes Medical Institute | Freeman Hrabowski | Claudia M Moreno |

The funders had no role in study design, data collection and interpretation, or the decision to submit the work for publication.

### Author contributions

Roya Pournejati, Conceptualization, Data curation, Formal analysis, Validation, Investigation, Writing – original draft, Writing – review and editing; Jessica M Huang, Resources, Formal analysis, Writing – original draft, Software, Writing – review and editing; Michael Ma, Methodology, Writing – review and editing; Claudia M Moreno, Conceptualization, Funding acquisition, Writing – review and editing; Oscar Vivas, Conceptualization, Formal analysis, Supervision, Funding acquisition, Investigation, Writing – original draft, Writing – review and editing

### Author ORCIDs

Roya Pournejati (iD) https://orcid.org/0000-0001-9290-4960
Jessica M Huang (iD) https://orcid.org/0009-0003-6452-3054
Claudia M Moreno (iD) https://orcid.org/0000-0001-8397-3649
Oscar Vivas (iD) https://orcid.org/0000-0002-0964-385X

Reviewer #1 (Public review): https://doi.org/10.7554/eLife.106791.4.sa1
Reviewer #3 (Public review): https://doi.org/10.7554/eLife.106791.4.sa2
Author response https://doi.org/10.7554/eLife.106791.4.sa3

## Additional files

### Supplementary files

MDAR checklist

Source code 1. Text code of the analysis program SpotScramble in PDF.

### Data availability

All datasets supporting the findings of this study have been deposited in Dryad and are publicly available in dryad https://doi.org/10.5061/dryad.63xsj3vfq.

The following dataset was generated:

| Author(s) | Year | Dataset title | Dataset URL | Database and Identifier |
| --- | --- | --- | --- | --- |
| Pournejati R, Huang JM, Ma M, Moreno C, Vivas O | 2025 | Functionally-coupled ion channels begin co-assembling at the start of their synthesis | https://doi.org/10.5061/dryad.63xsj3vfq | Dryad Digital Repository, 10.5061/dryad.63xsj3vfq |

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
