## [Editor Report · eLife Assessment]

This **fundamental** manuscript provides **compelling** evidence that BK and CaV1.3 channels can co-localize as ensembles early in the biosynthetic pathway, including within the ER and Golgi. The findings, supported by a range of imaging and proximity assays, offer insights into channel organization in both heterologous and endogenous systems. The data substantiate the central claims, while highlighting intriguing mechanistic questions for future studies: the determinants of mRNA co-localization, the temporal dynamics of ensemble trafficking, and the physiological implications of pre-assembly for channel function at the plasma membrane.

---

## [Referee Report · Reviewer #1 (Public review)]

Summary:

The co-localization of large conductance calcium- and voltage activated potassium (BK) channels with voltage-gated calcium channels (CaV) at the plasma membrane is important for the functional role of these channels in controlling cell excitability and physiology in a variety of systems. An important question in the field is where and how do BK and CaV channels assemble as 'ensembles' to allow this coordinated regulation - is this through preassembly early in the biosynthetic pathway, during trafficking to the cell surface or once channels are integrated into the plasma membrane. These questions also have broader implications for assembly of other ion channel complexes. Using an imaging based approach, this paper addresses the spatial distribution of BK-CaV ensembles using both overexpression strategies in tsa201 and INS-1 cells and analysis of endogenous channels in INS-1 cells using proximity ligation and superesolution approaches. In addition, the authors analyse the spatial distribution of mRNAs encoding BK and Cav1.3. The key conclusion of the paper that BK and CaV1.3 are co-localised as ensembles intracellularly in the ER and Golgi is well supported by the evidence. The experiments and analysis are carefully performed and the findings are very well presented.

---

## [Referee Report · Reviewer #3 (Public review)]

Summary:

The authors present a clearly written and beautifully presented piece of work demonstrating clear evidence to support the idea that BK channels and Cav1.3 channels can co-assemble prior to their assertion in the plasma membrane.

Strengths:

The experimental records shown back up their hypotheses and the authors are to be congratulated for the large number of control experiments shown in the ms.

---

## [Author Response]

The following is the authors’ response to the previous reviews.

**Reviewer #1 (Public review):**
Summary:This manuscript by Pournejati et al investigates how BK (big potassium) channels and CaV1.3 (a subtype of voltage-gated calcium channels) become functionally coupled by exploring whether their ensembles form early-during synthesis and intracellular trafficking-rather than only after insertion into the plasma membrane. To this end, the authors use the PLA technique to assess the formation of ion channel associations in the different compartments (ER, Golgi or PM), single-molecule RNA in situ hybridization (RNAscope), and super-resolution microscopy.Strengths:The manuscript is well written and addresses an interesting question, combining a range of imaging techniques. The findings are generally well-presented and offer important insights into the spatial organization of ion channel complexes, both in heterologous and endogenous systems.Weaknesses:The authors have improved their manuscript after revisions, and some previous concerns have been addressed.Still, the main concern about this work is that the current experiments do not quantitatively or mechanistically link the ensembles observed intracellularly (in the endoplasmic reticulum (ER) or Golgi) to those found at the plasma membrane (PM). As a result, it is difficult to fully integrate the findings into a coherent model of trafficking. Specifically, the manuscript does not address what proportion of ensembles detected at the PM originated in the ER. Without data on the turnover or halflife of these ensembles at the PM, it remains unclear how many persist through trafficking versus forming de novo at the membrane. The authors report the percentage of PLApositive ensembles localized to various compartments, but this only reflects the distribution of pre-formed ensembles. What remains unknown is the proportion of total BK and Ca_V_1.3 channels (not just those in ensembles) that are engaged in these complexes within each compartment. Without this, it is difficult to determine whether ensembles form in the ER and are then trafficked to the PM, or if independent ensemble formation also occurs at the membrane. To support the model of intracellular assembly followed by coordinated trafficking, it would be important to quantify the fraction of the total channel population that exists as ensembles in each compartment. A comparable ensemble-to-total ratio across ER and PM would strengthen the argument for directed trafficking of pre-assembled channel complexes.

We appreciate the reviewer’s thoughtful comment and agree that quantitatively linking intracellular hetero-clusters to those at the plasma membrane is an important and unresolved question. Our current study does not determine what proportion of ensembles at the plasma membrane originated during trafficking. It also does not quantify the fraction of total BK and Ca_V_1.3 channels engaged in these complexes within each compartment. Addressing this requires simultaneous measurement of multiple parameters—total BK channels, total Ca_V_1.3 channels, hetero-cluster formation (via PLA), and compartment identity—in the same cell. This is technically challenging. The antibodies used for channel detection are also required for the proximity ligation assay, which makes these measurements incompatible within a single experiment.

To overcome these limitations, we are developing new genetically encoded tools to enable real-time tracking of BK and Ca_V_1.3 dynamics in live cells. These approaches will enable us to monitor channel trafficking and the formation of hetero-clusters, as detected by colocalization. This kind of experiments will provide insight into their origin and turnover. While these experiments are beyond the scope of the current study, the findings in our current manuscript provide the first direct evidence that BK and CaV channels can form hetero-clusters intracellularly prior to reaching the plasma membrane. This mechanistic insight reveals a previously unrecognized step in channel organization and lays the foundation for future work aimed at quantifying ensemble-to-total ratios and determining whether coordinated trafficking of pre-assembled complexes occurs.

This limitation is acknowledged in the discussion section, page 23. It reads: “Our findings highlight the intracellular assembly of BK-Ca_V_1.3 hetero-clusters, though limitations in resolution and organelle-specific analysis prevent precise quantification of the proportion of intracellular complexes that ultimately persist on the cell surface.”

**Reviewer #2 (Public review):**
Summary:The co-localization of large conductance calcium- and voltage activated potassium (BK) channels with voltage-gated calcium channels (CaV) at the plasma membrane is important for the functional role of these channels in controlling cell excitability and physiology in a variety of systems.An important question in the field is where and how do BK and CaV channels assemble as 'ensembles' to allow this coordinated regulation - is this through preassembly early in the biosynthetic pathway, during trafficking to the cell surface or once channels are integrated into the plasma membrane. These questions also have broader implications for assembly of other ion channel complexesUsing an imaging based approach, this paper addresses the spatial distribution of BKCaV ensembles using both overexpression strategies in tsa201 and INS-1 cells and analysis of endogenous channels in INS-1 cells using proximity ligation and superesolution approaches. In addition, the authors analyse the spatial distribution of mRNAs encoding BK and Cav1.3.The key conclusion of the paper that BK and Ca_V_1.3 are co-localised as ensembles intracellularly in the ER and Golgi is well supported by the evidence.However, whether they are preferentially co-translated at the ER, requires further work. Moreover, whether intracellular pre-assembly of BK-Ca_V_1.3 complexes is the major mechanism for functional complexes at the plasma membrane in these models requires more definitive evidence including both refinement of analysis of current data as well as potentially additional experiments.

The reviewer raises the question of whether BK and Ca_V_1.3 channels are preferentially co-translated. In fact, I would like to propose that co-translation has not yet been clearly defined for this type of interaction between ion channels. In our current work, we (1) observed the colocalization between BK and Ca_V_1.3 mRNAs and (2) determined that 70% of BK mRNA in active translation also colocalizes with Ca_V_1.3 mRNA. We think these results favor the idea of translational complexes that can underlie the process of co-translation. However, and in total agreement with the Reviewer, the conclusion that the mRNA for the two ion channels is cotranslated would require further experimentation. For instance, mRNA coregulation is one aspect that could help to define co-translation.

To avoid overinterpretation, we have revised the manuscript to remove references to “co-translation” in the Results section and included the word “potential” when referring to co-translation in the Discussion section. We also clarified the limitations of our evidence in the Discussion that can be found on page 25: “It is important to note that while our data suggest mRNA coordination, additional experiments are required to directly assess co-translation.”

Strengths & Weaknesses(1) Using proximity ligation assays of overexpressed BK and CaV1.3 in tsa201 and INS1 cells the authors provide strong evidence that BK and CaV can exist as ensembles (ie channels within 40 nm) at both the plasma membrane and intracellular membranes, including ER and Golgi. They also provide evidence for endogenous ensemble assembly at the Golgi in INS-1 cells and it would have been useful to determine if endogenous complexes are also observe in the ER of INS-1 cells. There are some useful controls but the specificity of ensemble formation would be better determined using other transmembrane proteins rather than peripheral proteins (eg Golgi 58K).

We thank the reviewer for their thoughtful feedback and for recognizing the strength of our proximity ligation assay data supporting BK–Ca_V_1.3 hetero-clusters formation at both the plasma membrane and intracellular compartments. As for specificity controls, we appreciate the suggestion to use transmembrane markers. To strengthen our conclusion, we have performed an additional experiment comparing the number of PLA puncta formed by the interaction of Ca_V_1.3 and BK channels with the number of PLA puncta formed by the interaction of Ca_V_1.3 channels and ryanodine receptors in INS-1 cells. As shown in the figure below, the number of interactions between Ca_V_1.3 and BK channels is significantly higher than that between Ca_V_1.3 and RyR_2_. Of note, RyR_2_ is a protein resident of the ER. These results provide additional evidence of the existence of endogenous complex formation in INS-1 cells. We have added this figure as a supplement.

(2) Ensemble assembly was also analysed using super-resolution (dSTORM) imaging in INS-1 cells. In these cells only 7.5% of BK and CaV particles (endogenous?) co-localise that was only marginally above chance based on scrambled images. More detailed quantification and validation of potential 'ensembles' needs to be made for example by exploring nearest neighbour characteristics (but see point 4 below) to define proportion of ensembles versus clusters of BK or Cav1.3 channels alone etc. For example, it is mentioned that a distribution of distances between BK and Cav is seen but data are not shown.

We thank the reviewer for this comment. To address the request for more detailed quantification and validation of ensembles, we performed additional analyses:

Proportion of ensembles vs isolated clusters: We quantified clusters within 200 nm and found that 37 ± 3% of BK clusters are near one or more CaV1.3 clusters, whereas 15 ± 2% of CaV1.3 clusters are near BK clusters. Figure 8– Supplementary 1A

Distance distribution: As shown in Figure 8–Supplementary 1B, the nearestneighbor distance distribution for BK-to-CaV1.3 in INS-1 cells (magenta) is shifted toward shorter distances compared to randomized controls (gray), supporting preferential localization of BK–CaV1.3 hetero-clusters.

Together, these analyses confirm that BK–CaV1.3 ensembles occur more frequently than expected by chance and exhibit an asymmetric organization favoring BK proximity to CaV1.3 in INS-1 cells. We have included these data and figures in the revised manuscript, as well as description in the Results section.

(3) The evidence that the intracellular ensemble formation is in large part driven by cotranslation, based on co-localisation of mRNAs using RNAscope, requires additional critical controls and analysis. The authors now include data of co-localised BK protein that is suggestive but does not show co-translation. Secondly, while they have improved the description of some controls mRNA co-localisation needs to be measured in both directions (eg BK - SCN9A as well as SCN9A to BK) especially if the mRNAs are expressed at very different levels. The relative expression levels need to be clearly defined in the paper. Authors also use a randomized image of BK mRNA to show specificity of co-localisation with Cav1.3 mRNA, however the mRNA distribution would not be expected to be random across the cell but constrained by ER morphology if cotranslated so using ER labelling as a mask would be useful?

We thank the reviewer for these constructive suggestions. We measured mRNA colocalization in both directions as recommended. As shown in the figure below, colocalization between KCNMA1 and SCN9A transcripts was comparable in both directions, with no statistically significant difference, supporting the specificity of the observed associations. We decided not to add this to the original figure to keep the figure simple.

We agree that co-localization of BK protein with BK mRNA is not conclusive evidence of co-translation, and we do not intend to mislead readers in our conclusion. Consequently, we were careful in avoiding the use of co-translation in the result section and added the word “potential” when referring to co-translation in the Discussion section. We added a sentence in the discussion to caution our interpretation: “It is important to note that while our data suggest mRNA coordination, additional experiments are required to directly assess cotranslation.”

(4) The authors attempt to define if plasma membrane assemblies of BK and CaV occur soon after synthesis. However, because the expression of BK and CaV occur at different times after transient transfection of plasmids more definitive experiments are required. For example, using inducible constructs to allow precise and synchronised timing of transcription. This would also provide critical evidence that co-assembly occurs very early in synthesis pathways - ie detecting complexes at ER before any complexes

We appreciate the reviewer’s insightful suggestion regarding the use of inducible constructs to synchronize transcription timing. This is an excellent approach and would allow direct testing of whether co-assembly occurs early in the synthesis pathway, including detection of complexes at the ER prior to plasma membrane localization. These experiments are beyond the scope of the present work but represent an important direction for future studies.

We have added the following sentence to the Discussion section (page 24) to highlight this idea. “Future experiments using inducible constructs to precisely control transcription timing will enable more precise quantification of heterocluster formation in the ER compartment prior to plasma membrane insertion and reduce the variability introduced by differences in expression timing after plasmid transfection.”

(5) While the authors have improved the definition of hetero-clusters etc it is still not clear in superesolution analysis, how they separate a BK tetramer from a cluster of BK tetramers with the monoclonal antibody employed ie each BK channel will have 4 binding sites (4 subunits in tetramer) whereas Cav1.3 has one binding site per channel. Thus, how do authors discriminate between a single BK tetramer (molecular cluster) with potential 4 antibodies bound compared to a cluster of 4 independent BK channels.

We appreciate the reviewer’s thoughtful comment regarding the interpretation of super-resolution data. We agree that distinguishing a single BK tetramer from a cluster of multiple BK channels is challenging when using an antibody that can bind up to four sites per channel. To clarify, our analysis does not attempt to resolve individual subunits within a tetramer; rather, it focuses on the nanoscale spatial proximity of BK and Ca_V_1.3 signals.

We want to note that this limitation applies only to the super-resolution maps in Figures 8C and 9D and does not affect Airyscan-based analyses or measurements of BK–Ca_V_1.3 proximity.

To address how we might distinguish between a single BK tetramer and a cluster of multiple BK channels, we considered two contrasting scenarios. In the first case, we assume that all four α-subunits within a tetramer are labeled. Based on cryoEM structures, a BK tetramer measures approximately 13 nm × 13 nm (≈169 nm²). Adding two antibody layers (primary and secondary) would increase the footprint by ~14 nm in each direction, resulting in an estimated area of ~41 nm × 41 nm (≈1681 nm²). Under this assumption, particles smaller than ~1681 nm² would likely represent individual tetramers, whereas larger particles would correspond to clusters of multiple tetramers.

In the second scenario, we propose that steric constraints at the S9–S10 segment, where the antibody binds, limit labeling to a single antibody per tetramer. If true, the localization precision would approximate 14 nm × 14 nm—the combined size of the antibody complex and the channel—close to the resolution limit of the microscope. To test this, we performed a control experiment using two antibodies targeting the BK C-terminal domain, raised in different species and labeled with distinct fluorophores. Super-resolution imaging revealed that only ~12% of particles were colocalized, suggesting that most channels bind a single antibody.

If multiple antibodies could bind each tetramer, we would expect much greater colocalization.

Although these data are not included in the manuscript, we have added the following clarification to the Results section (page 19): “It is important to note that this technique does not allow us to distinguish between labeling of four BK αsubunits within a tetramer and labeling of multiple BK channel clusters. Hence, particles smaller than ~1680 nm² may represent either a single tetramer or a cluster. This limitation applies to Figures 8C and 9D and does not affect measurements of BK–Ca_V_1.3 proximity.”

**Author response image 2. sa3fig2:** 

(6) The post-hoc tests used for one way ANOVA and ANOVA statistics need to be defined throughout

We thank the reviewer for highlighting the need for clarity regarding our statistical analyses. We have now specified the post-hoc tests used for all one-way ANOVA and ANOVA comparisons throughout the manuscript, and updated figure legends.

**Reviewer #3 (Public review):**
Summary:The authors present a clearly written and beautifully presented piece of work demonstrating clear evidence to support the idea that BK channels and Cav1.3 channels can co-assemble prior to their assertion in the plasma membrane.Strengths:The experimental records shown back up their hypotheses and the authors are to be congratulated for the large number of control experiments shown in the ms.
**Recommendations for the authors:**

**Reviewer #1 (Recommendations for the authors):**
The authors have sufficiently addressed the specific points previously raised and the manuscript has improved clarity in those aspects. My main concern, which still remains, is stated in the public review.
**Reviewer #3 (Recommendations for the authors):**
I am content that the authors have attempted to fully address my previous criticisms.I have only three suggestions(1) I think the word Homo-clusters at the bottom right of Figure 1 is erroneously included.

We thank the reviewer for bringing this to our attention. The figure has been corrected accordingly.

(2) The authors should, for completeness, to refer to the beta, gamma and LINGO subunit families in the Introduction and include appropriate references:Knaus, H. G., Folander, K., Garcia-Calvo, M., Garcia, M. L., Kaczorowski, G. J., Smith, M., & Swanson, R. (1994). Primary sequence and immunological characterization of betasubunit of high conductance Ca2+-activated K+ channel from smooth muscle. The Journal of Biological Chemistry, 269(25), 17274-17278.Brenner, R., Jegla, T. J., Wickenden, A., Liu, Y., & Aldrich, R. W. (2000a). Cloning and functional characterization of novel large conductance calcium-activated potassium channel beta subunits, hKCNMB3 and hKCNMB4. The Journal of Biological Chemistry, 275(9), 6453-6461.Yan, J & R.W. Aldrich. (2010) LRRC26 auxiliary protein allows BK channel activation at resting voltage without calcium. Nature. 466(7305):513-516Yan, J & R.W. Aldrich. (2012) BK potassium channel modulation by leucine-rich repeatcontaining proteins. Proceedings of the National Academy of Sciences 109(20):7917-22Dudem, S, Large RJ, Kulkarni S, McClafferty H, Tikhonova IG, Sergeant, GP, Thornbury, KD, Shipston, MJ, Perrino BA & Hollywood MA (2020). LINGO1 is a novel regulatory subunit of large conductance, Ca2+-activated potassium channels. Proceedings of the National Academy of Sciences 117 (4) 2194-2200Dudem, S., Boon, P. X., Mullins, N., McClafferty, H., Shipston, M. J., Wilkinson, R. D. A., Lobb, I., Sergeant, G. P., Thornbury, K. D., Tikhonova, I. G., & Hollywood, M. A. (2023). Oxidation modulates LINGO2-induced inactivation of large conductance, Ca2+-activated potassium channels. The Journal of Biological Chemistry, 299 (3) 102975.

We agree with the reviewer’s suggestion and have revised the Introduction to include references to the beta, gamma, and LINGO subunit families. Appropriate citations have been added to ensure completeness and contextual relevance.

Additionally, BK channels are modulated by auxiliary subunits, which fine-tune BK channel gating properties to adapt to different physiological conditions. The β, γ, and LINGO1 subunits each contribute distinct structural and regulatory features: β-subunits modulate Ca²⁺ sensitivity and can induce inactivation; γ-subunits shift voltage-dependent activation to more negative potentials; and LINGO1 reduces surface expression and promotes rapid inactivation (18-24). These interactions ensure precise control over channel activity, allowing BK channels to integrate voltage and calcium signals dynamically in various cell types.

(3) I think it may be more appropriate to include the sentence "The probes against the mRNAs of interest and tested in this work were designed by Advanced Cell Diagnostics." (P16, right hand column, L12-14) in the appropriate section of the Methods, rather than in Results.

We thank the reviewer for this helpful suggestion. In response, we have relocated the sentence to the appropriate section of the Methods, where it now appears with relevant context.